# From small scale variability to mesoscale stability in surface
# ocean pH: implications for air-sea CO₂ equilibration
Louise Delaigue[1,2], Gert-Jan Reichart[1,3], Li Qiu[4], Eric P. Achterberg[4], Yasmina Ourradi[1], Chris
Galley[5,6], André Mutzberg[4] and Matthew P. Humphreys[1]
[1]Department of Ocean Systems (OCS), NIOZ Royal Netherlands Institute for Sea Research, PO Box 59, 1790 AB
Den Burg (Texel), the Netherlands
[2]Sorbonne Université, CNRS, Laboratoire d'Océanographie de Villefranche, LOV, 06230 Villefranche-sur-Mer,
France
[3]Department of Earth Sciences, Utrecht University, 3584 CS Utrecht, The Netherlands
[4]GEOMAR Helmholtz Centre for Ocean Research, Kiel, Germany
[5]Department of Earth Sciences, Memorial University of Newfoundland, St. John's, NL, A1B 3X5, Canada
[6]Department of Earth and Environmental Sciences, University of Ottawa, Ottawa, ON, K1N 6N5, Canada
*Correspondence to*: Louise Delaigue (louise.delaigue@imev-mer.fr)
**Abstract**. One important aspect of understanding ocean acidification is the nature and drivers of pH variability in
surface waters on smaller spatial (i.e., areas up to 100 km²) and temporal (i.e., days) scales. However, there has been
a lack of high-quality pH data at sufficiently high resolution. Here, we describe a simple optical system for continuous
high-resolution surface seawater pH measurements. The system includes a PyroScience pH optode placed in a flow-
through cell directly connected to the underway supply of a ship through which near-surface seawater is constantly
pumped. Seawater pH is measured at a rate of 2 to 4 measurements min⁻¹ and is cross-calibrated using discrete
carbonate system observations (total alkalinity, dissolved inorganic carbon, and nutrients). This setup was used during
two research cruises in different oceanographic conditions: the North Atlantic Ocean (December 2020-January 2021)
and the South Pacific Ocean (February-April 2022). By leveraging this novel high-frequency measurement approach,
our findings reveal fine-scale fluctuations in surface seawater pH across the North Atlantic and South Pacific Oceans.
While temperature is a significant abiotic factor driving these variations, it does not account for all observed changes.
Instead, our results highlight the interplay between temperature, biological activity, and waters with distinct
temperature-salinity properties on pH. Notably, the variability differed between the two regions, suggesting
differences in the dominant factors influencing pH. In the South Pacific, biological processes appeared to be mostly
responsible for pH variability, while in the North Atlantic, additional abiotic and biotic factors complicated the
correlation between expected and observed pH changes. While our findings indicate that broader ocean-basin scale
analyses based on lower-resolution datasets can effectively capture surface ocean CO₂ variability at a global scale,
they also highlight the necessity of fine-scale observations for resolving regional processes and their drivers, which is
essential for improving predictive models of ocean acidification and air-sea CO₂ exchange.
## 1 Introduction
Ocean chemistry is changing due to the uptake of anthropogenic CO₂ from the atmosphere
(DeVries, 2022). The uptake of atmospheric CO₂ by the ocean's surface increases hydrogen ion
concentration, a process known as ocean acidification, which has led to a 30–40% rise in hydrogen
ion concentration (i.e. surface seawater acidity) and a corresponding pH decrease of ~0.1 since
around 1850 (Gattuso et al., 2015; Jiang et al., 2019; Orr et al., 2005). These changes have already
significantly impacted marine organisms, especially marine calcifiers (Doney et al., 2020; Gattuso
et al., 2015; Osborne et al., 2020), and pH is projected to decline by ~0.3 by 2100 (Figuerola et al.,
41  2021).


High-resolution studies of surface ocean carbonate chemistry and air-sea CO₂ exchange have
significantly advanced our understanding of the upper ocean's carbon cycle. However, gaps
remain, particularly at fine spatio-temporal scales (e.g., variability over hours and a few
kilometers). At these scales, changes in pH over short time periods can be an important control on
the ocean's buffering capacity and response to $CO_2$ uptake, highlighting the need for further
detailed observations (Cornwall et al., 2013; Egilsdottir et al., 2013; James et al., 2020; Qu et al.,
2017; Wei et al., 2022).
Advancements in the last decade and a half have enhanced the capacity for accurate and precise
in-situ pH measurements. For example, Martz et al. (2010) developed an autonomous sensor
tailored for continuous deployment in marine environments that allows recording of high-
resolution pH fluctuations. Autonomous surface vehicles in coastal upwelling systems have also
been used to capture intricate partial pressure of $CO_2$ ($pCO_2$) and pH dynamics, even in complex
environments where rapid biogeochemical changes occur due to natural phenomena like upwelling
(Chavez et al., 2018; Cryer et al., 2020; Possenti et al., 2021). Staudinger et al. (2018) also
developed an optode system capable of simultaneously measuring oxygen, carbon dioxide, and pH
in seawater. This system was designed for extended deployment (i.e. days) in marine
environments, enabling continuous monitoring (with measurement intervals between 1 second and
1 hour) of these parameters. Additionally, Sutton et al. (2019) detailed the implementation of
autonomous seawater $pCO_2$ and pH time series from 40 surface buoys, broadening the scope of
observations at fixed time series sites. Staudinger et al. (2019) introduced fast and stable optical
pH sensor materials specifically for oceanographic applications, enhancing the ability to measure
pH under various environmental conditions.
These technological advancements have facilitated significant scientific progress. Field
measurements conducted using submersible spectrophotometric sensors have revealed fine-scale
variations in pH in coastal waters and shed light on localized acidification processes (Cornwall et
al., 2013). The implementation of autonomous seawater $pCO_2$ and pH time series as described by
Sutton et al. (2019) has enhanced our ability to characterize sub-seasonal variability in the ocean.
These efforts represent important progress that can be built upon to further understand fine-scale
ocean dynamics.
Fine spatio-temporal scale variability in surface ocean pH is hard to capture because it is driven
by a complex interplay of processes, including physical mixing, biological activity (i.e.
photosynthesis and respiration), thermal variability, and air-sea $CO_2$ fluxes (Faassen et al., 2023;
Hofmann et al., 2011; Price et al., 2012). Physical mixing moderates surface oceanic pH by
redistributing dissolved $CO_2$, nutrients, and heat throughout the water column. Mixing also
mitigates extreme pH fluctuations by diluting surface concentrations of $CO_2$ during periods of high
biological activity or temperature-induced $CO_2$ release (Egea et al., 2018; Li et al., 2019).
Photosynthetic activity can decrease $CO_2$, leading to an increase in pH during daylight hours, while
respiration dominates at night, releasing $CO_2$ and lowering pH (Fujii et al., 2021; Jokiel et al.,
2014). Warmer waters decrease $CO_2$ solubility and increase pH, while cooler waters increase
solubility, promoting $CO_2$ uptake and decreasing pH, although the timescale of these processes
differs, with some changes occurring instantaneously and others after equilibration (Zeebe &
Wolf-Gladrow, 2001). Instantaneous changes are driven by physical and chemical reactions, while
equilibration processes involve longer-term adjustments such as air-sea gas exchange and the
mixing of surface waters with deeper layers (Emerson & Hedges, 2008). When atmospheric $CO_2$
exceeds oceanic $CO_2$, the ocean takes up $CO_2$, lowering pH; conversely, when atmospheric $CO_2$
decreases below oceanic $CO_2$, outgassing occurs, raising pH (Caldeira & Wickett, 2005; Orr et al.,
2005). Although each of these processes has its distinct impact on pH, their combined effects
regulate the ocean's carbon cycle and its interaction with the atmosphere.
Recent studies on air-sea $CO_2$ equilibration timescales have highlighted significant regional
variations, particularly between the North Atlantic and South Pacific Oceans (Jones et al., 2014).
In the North Atlantic, equilibration timescales for $CO_2$ between the atmosphere and the ocean's
surface mixed layer vary significantly with latitude. In regions above 55°N, these timescales can
extend up to 18 months, while at lower latitudes, such as around 30°N, they range from 3 to 6
months (Jones et al., 2014). These long equilibration timescales reduce the extent to which the
ocean can buffer short-term changes in surface pH. On pentadal and longer timescales, the air-sea
$CO_2$ flux in the North Atlantic is driven primarily by changes in $\Delta pCO_2$, while gas transfer velocity
plays a more significant role only on interannual and shorter timescales (Couldrey et al., 2016).
Cooler temperatures at higher latitudes increase $CO_2$ solubility, resulting in higher dissolved
inorganic carbon (DIC), and upwelling brings DIC and total alkalinity (TA)-rich deep waters to
the surface (Wu et al., 2019). These factors further increase the amount of $CO_2$ that needs to
exchange with the atmosphere, further prolonging equilibration times. The South Pacific, with its
shallower mixed layers and higher average surface temperatures, facilitates shorter equilibration
times and enhances $CO_2$ uptake rates (i.e. 3 to 4 months; Jones et al., 2014). Wu et al. (2019) also
showed that high biological productivity in these areas significantly impacts DIC, potentially
reducing surface DIC more quickly.
Here, we test a high-frequency optical system to investigate how surface seawater pH varies across
fine spatio-temporal scales, focusing on changes occurring over areas up to 100 $km^2$ and timescales
of hours to days across different ocean basins (i.e., North Atlantic and South Pacific Oceans) and
identify abiotic and biotic factors driving these variations. We use direct, high-frequency
measurements of surface seawater pH and estimate TA to resolve the rest of the carbonate system.
This novel setup provides new insights into processes, such as temperature, hydrodynamic mixing
and biological activity, influence fine-scale spatio-temporal variability in pH.
**2 Materials and Procedures**
**2.1 Study areas**

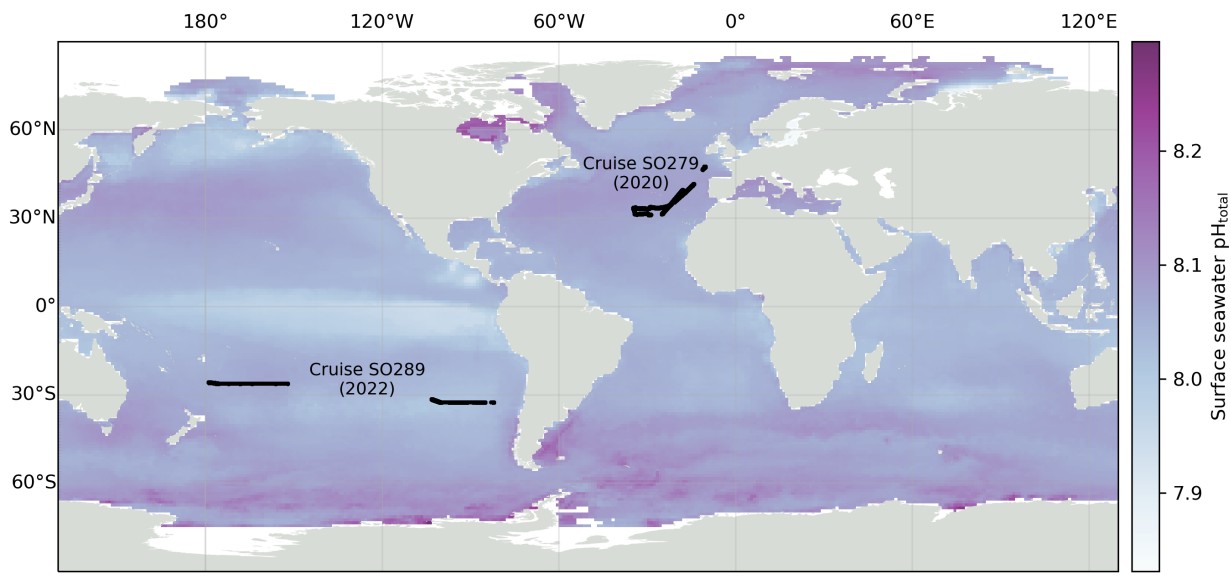

**Figure 1. Locations of pH measurements during two oceanographic cruises used in this study: SO279 in the**
**North Atlantic (December 2020 to January 2021) and SO289 in the South Pacific (February-April 2022).**
**Surface seawater pH on the total scale for December 2022 from the OceanSODA product is shown in the**
**background (**Gregor & Gruber, 2021**).**
Two datasets from separate oceanic regions were used: research expeditions SO279 in the North
Atlantic Ocean and SO289 in the South Pacific Ocean, both on the German R/V Sonne (Fig. 1).
Expedition SO279 in January to December 2020, conducted in the Azores Region of the North
Atlantic Ocean (Fig. 1), was part of the NAPTRAM research program investigating the transport
pathways of plastic and microplastic debris (Beck et al., 2021). The data collection included
discrete samples from the CTD rosette (n=77; Delaigue et al., 2021a), with measurements of DIC,
TA and nutrients (silicate, phosphate ammonium, nitrite, and nitrate + nitrite); discrete samples
from the underway water system (UWS; n=51; Delaigue et al., 2021b) also measuring these
parameters; and a high-resolution UWS timeseries of ocean surface pH (over 43,000 datapoints;
Delaigue et al., 2021c).
The South Pacific GEOTRACES Cruise SO289, from Valparaiso (Chile) to Noumea (New
Caledonia) under the GEOTRACES GP21 initiative, was conducted from February to April 2022
(Fig. 1; Achterberg et al., 2022). The data collection also included the same parameters as SO279,
with discrete samples from the CTD rosette (n=395; Delaigue, Ourradi, Ossebar, et al., 2023),
discrete samples from the UWS (n=32; Delaigue, Ourradi, et al., 2023a) and another high-
resolution UWS timeseries of ocean surface pH from the optode system (over 78,000 datapoints;
Delaigue, Ourradi, et al., 2023b).
**2.2 Integrated shipboard optode system for continuous pH measurements**
We used a pH optode (PHROBSC-PK8T, for pH range 7.0 – 9.0 on the total scale; PyroScience
GmbH), made of a robust cap adapter fiber with a stainless-steel tip (length=10cm,
diameter=4mm) and disposable plastic screw cap with an integrated pH sensor. The total scale
accounts for sulfate ion dissociation in seawater, providing a more accurate representation of
carbonate system equilibria compared to other pH scales commonly used in marine chemistry.
Unless explicitly stated otherwise, all references to pH in this manuscript refer to pH on the total
scale. The manufacturer-reported accuracy of the optode is ±0.05 for pH 7.5–9.0 and ±0.1 for pH
7.0–7.5 after a two-point calibration, with a precision of ±0.003 at pH 8.0.

The optode was connected to a meter combined with a pressure-stable optical connector (optical
feed-through; OEM module Pico-pH-SUB; PyroScience GmbH; Fig. 2). Briefly, the optical pH
sensor is constructed using the PyroScience REDFLASH technology, which uses a pH neutral
reference indicator and a pH responsive luminescent dye. These elements are activated using a
specifically tuned orange-red light with a wavelength ranging between 610-630 nm, which triggers
a bright luminescence emission in the near-infrared (NIR) band, spanning from 760-790 nm. At
elevated pH, the fluorescence from the pH marker is diminished, leaving only the NIR emission
of the reference indicator noticeable. As the acidity increases, the pH marker is protonated, which
results in a heightened NIR luminescence that is detected along with the emissions of both
indicators. The measurement approach uses red excitation light modulated in a sinusoidal manner,
leading to a similar modulation of the NIR emission, albeit with a phase discrepancy. This phase
variation is registered by the PyroScience OEM module and subsequently converted into a total
pH measurement.

Automatic temperature compensation of the optical pH sensor was achieved using a flexible
Teflon-coated temperature probe (Pt100 Temperature Probe, PyroScience GmbH; Fig. 2) soldered
onto the OEM module. The optode was placed in a closed flow-through cell directly connected to
the underway supply of the ship through which seawater was pumped at a constant rate (6 L/min
for SO279 and 9 L/min for SO289; 2.5 m depth) and stirred using a magnetic stirrer (Fig. 2). The
entire setup was kept inside a closed box to isolate the optical instrument from any other light
source (Fig. 2). All seawater first went through a thermo-salinograph close to the water source,
which also measured salinity, temperature and chlorophyll-a fluorescence (chl-a), before going
through the pH setup (Fig. 2).

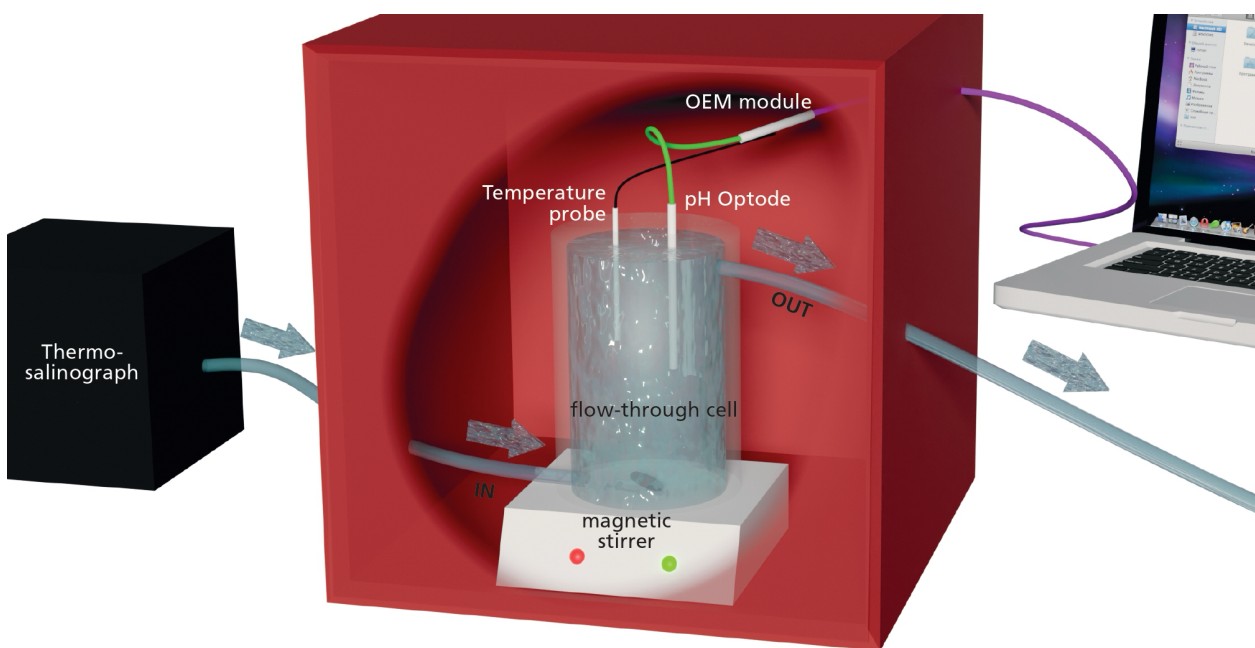

**Figure 2. Schematic representation of the optical continuous pH measurement system. Arrows indicate the direction of flow-through tubing. The system consists of the following components: a thermo-salinograph; a flow-through cell; a magnetic stirrer; a fiber-based pH optode; a flexible Teflon-coated temperature probe; a fiber-optic meter OEM module Pico-pH-SUB and a portable computer. All elements inside the red box are in the dark to avoid any light disturbance.**

### 2.3 Initial calibration and underway measurements

Direct measurements of surface water (~3 m depth) pH were carried out at a frequency of 2 measurements min$^{-1}$ for the North Atlantic cruise and 4 measurements min$^{-1}$ for the South Pacific cruise.

A one-point calibration of the temperature probe was performed against a thermometer inside a water bath (Lauda Ecoline RE106). A two-point calibration of the pH sensor was conducted following the manufacturer's recommended procedure, using PyroScience pH buffer capsules (pH 2 or pH 4 for the acidic calibration point, pH 10 or pH 11 for the basic calibration points). These calibration points deliberately fall far outside the sensor's operating range from pH 7-9 to characterize its maximum and minimum responses. Buffers were prepared by dissolving each capsule's powder into 100 mL MilliQ water.

To further refine the accuracy of the measurements, a pH offset adjustment was applied using certified reference material (CRM, batches #189, #195, and #198; provided by Andrew Dickson, Scripps Institution of Oceanography). Although CRMs do not provide direct certified pH values, we calculated pH$_{CRM}$ from the CRM TA and DIC using the carbonate system equilibrium constants from Lueker et al. (2000). This pH$_{CRM}$ value was then used to adjust the optode-based pH measurements to improve accuracy and align them with discrete observations.
To assess the robustness of the pH optode, resolution surface measurements of partial pressure of CO$_2$ (pCO$_2$) and pH (samipH) were added as points of comparison. The pH optode, pCO$_2$ sensor, and samipH sensor were all installed on the same surface seawater supply, which was continuously pumped from approximately 2.5 m depth into the sink of the underway laboratory during the

cruise. The PyroScience pH sensor received water from a split of this same supply, ensuring all
sensors were sampling the same source water.
Surface $pCO_2$ was measured using a factory-calibrated commercial sensor (HydroC, 4H-JENA
Engineering GmbH, Germany; Fietzek et al., 2014) at a 1-minute sampling interval. An automatic
zero-point calibration was performed every 6 hours to correct for sensor drift. The recorded $pCO_2$
values were subsequently adjusted following (Takahashi et al., 2006) to reflect in situ sea surface
temperature.
Surface pH, reported on the total scale, was measured using a factory-calibrated sensor (Sunburst
Sensors, USA) at 15-minute intervals. The pH data were calibrated using sea surface salinity and
temperature according to the equations provided by Liu et al. (2011) and Millero (2007),
respectively.

## 2.4 Discrete sampling and analysis for other $CO_2$ parameters

The underway seawater system was sub-sampled from the cell every 12 hours via silicone tubing
for TA and DIC following an internationally established protocol (Dickson et al., 2007). TA was
sampled in Azlon™ HDPE wide neck round 150 mL bottles filled to the neck and poisoned with
50 μL saturated $HgCl_2$. DIC was sampled into Labco Exetainer® 12 ml borosilicate vials and
poisoned with 15 μL saturated $HgCl_2$. Samples were stored at 4 °C whenever possible and kept in
the dark until analysis.
All TA and DIC analysis was carried out at the Royal Netherlands Institute for Sea Research, Texel
(NIOZ). The analysis was calibrated using certified reference material (CRM, batches #189, #195
and #198; provided by Andrew Dickson, Scripps Institution of Oceanography). TA was
determined using a Versatile INstrument for the Determination of Total inorganic carbon and
titration Alkalinity (VINDTA 3C #017 and #014, Marianda, Germany). The instrument performed
an open-cell, potentiometric titration of a seawater subsample with 0.1 M hydrochloric acid (HCl).
Results were then recalculated using a modified least-squares fitting as implemented by Calkulate
v3.1.0 (Humphreys and Matthews, 2024). DIC concentrations were determined using either the
VINDTA system (SO279 samples and part of SO289 samples) or the QuAAtro Gas Segmented
Continuous Flow Analyzer (CFA, SEAL Analytical; SO289 samples). Briefly, the VINDTA
measures DIC by acidifying a seawater sample, which releases $CO_2$ that is then quantified through
a coulometric titration cell. Similarly, the QuAAtro CFA uses acidification to liberate $CO_2$, which
then discolours a slightly alkaline phenolphthalein pink coloured solution which is measured
spectrophotometrically at 520 nm (Stoll et al., 2001).
For cruise SO279, nutrient samples were gathered using 60 mL syringes made of high-density
polyethylene, which were connected to a three-way valve by tubing, drawing directly from the
CTD-rosette bottles to avoid air exposure. Immediately upon collection, the samples were taken
to the laboratory for processing, where they were filtered through a dual-layer filter with pore sizes
of 0.8 and 0.2 μm. All samples were stored at -20°C in a freezer except Si, which were stored at
4°C in a cold room until analysis back at NIOZ. Nutrients were analysed using a QuAAtro
Continuous Flow Analyser. Measurements were made simultaneously on four channels together
Si, $PO_4$, $NH_4$, $NO_3$, and $NO_2$. All measurements were calibrated with standards diluted in low
nutrient seawater (LNSW) in the salinity range of the stations (approx. 34 – 37) to ensure that
analysis remained within the same ionic strength. Prior to analysis, all samples were brought to
laboratory temperature in about one to two hours. To avoid gas exchange and evaporation during
the runs with $NH_4$ analysis, all vials including the calibration standards were covered with Parafilm
under tension before being placed into the auto-sampler, so that the sharpened sample needle easily
penetrated through the film leaving only a small hole. Silicate samples were measured separately
on a TRAACS Gas Segmented Continuous Flow Analyser (manufactured by Bran+Lubbe, now
SEAL Analytical) following Strickland and Parsons (1972). A sampler rate of 60 samples per hour
was also used for all analyses. Calibration standards were diluted from stock solutions of the
different nutrients in 0.2 μm filtered LNSW diluted with de-ionised water to obtain approximately
the same salinity as the samples and were freshly prepared every day. This diluted LNSW was also
used as the baseline water for the analysis and in between the samples. Each run of the system
had a correlation coefficient of at least 0.9999 for 10 calibration points. The samples were
measured from the lowest to the highest concentration, i.e., from surface to deep waters in order
to reduce carry-over effects. Concentrations were recorded in μM at an average container
temperature of 23.0 °C and later converted to μmol/kg by dividing the recorded concentration by
the sample density, calculated following Millero and Poisson (1981).
For cruise SO289, nutrient analysis was carried out on seawater from every Niskin bottle triggered
at various depths during each cast. The seawater was transferred into 15 mL polypropylene vials.
Each container and its cap were rinsed three times with seawater before filling. If immediate
analysis was not possible, samples were stored in a fridge at 4 °C in the dark. Macro nutrients were
analyzed onboard using a segmented flow injection analysis with a Seal Analytical QUAATRO39
auto-analyzer that includes an XY2-autosampler. For nanomolar nutrient analysis, a modified
setup with 1000 mm flow cells was employed. The setup was designed to analyze four channels:
total oxidized nitrogen (TON), silicate, nitrite, and phosphate, using methods outlined in QuAAtro
Applications: Method Nos. Q-068-05 Rev. 11, Q-066-05 Rev. 5, Q-070-05 Rev. 6, and Q-064-05
Rev. 8, respectively. To ensure analytical consistency and validate the data, each run was checked
against Certified Reference Material for Nutrients in Seawater (RMNS). Nutrient analyses were
further validated using KANSO CRM, with specific lot numbers for macromolar and nanomolar
nutrient concentrations.

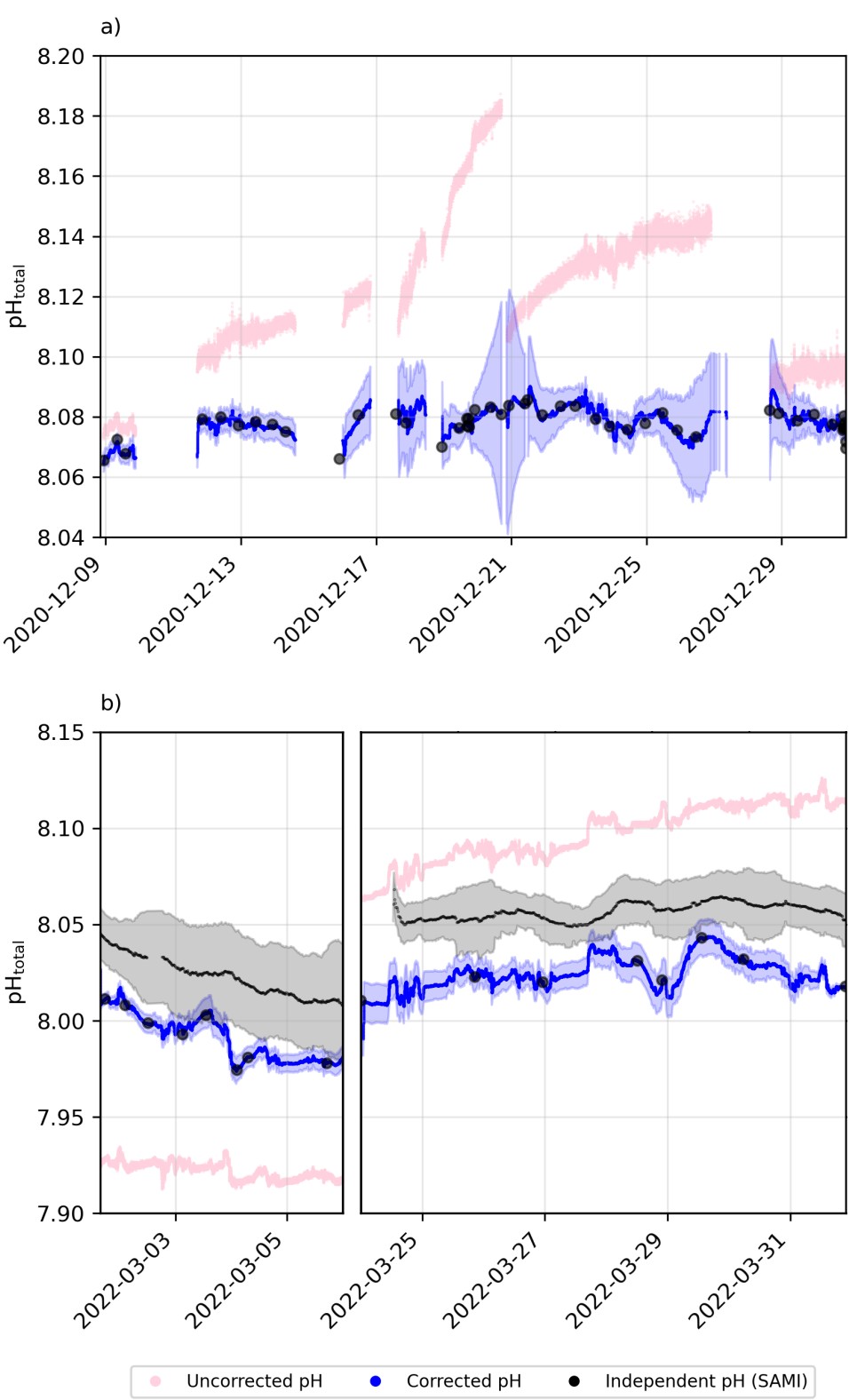

**Figure 3. Underway pH drift correction using pH$_{total(TA/DIC)}$ subsamples for a) cruise SO279 and b) cruise**
**SO289. Raw pH measurements (pink); corrected pH values and uncertainty from bootstrapping (blue);**
**subsample pH$_{total(TA/DIC)}$ depicted as black circles. Independent pH from the SAMI sensor is shown in black**
**for the South Pacific cruise (SO289).**

**2.5 Post-cruise correction**


To ensure the reliability of the dataset, an initial screening was conducted to identify and flag
unreliable continuous pH data points, primarily attributable to optode stabilization issues.
Specifically, data points recorded during documented periods of particularly rough weather, when
air intrusion into the underway system was reported, were flagged as unreliable due to the resulting
abrupt and erratic drift patterns inconsistent with expected optode and surface ocean pH behavior.

pH$_{obs}$ was calculated from TA and DIC UWS discrete samples) using PyCO2SYS (Version 1.8.2;
Humphreys et al., 2022)), with the carbonic acid dissociation constants of Lueker et al. (2000), the
bisulfate dissociation constant of Dickson (1990), the total boron to chlorinity ratio of Uppström
(1974), and the hydrogen fluoride dissociation constant of Dickson and Riley (1979).These
calculated values were then aligned with the continuous pH dataset to determine the offset between
pH$_{obs}$ subsamples and the continuous optode-based pH measurements (Fig. 3). A Piecewise Cubic
Hermite Interpolating Polynomial (PCHIP; Fritsch & Carlson, 1980) was fitted to the offset as a
way to continuously correct pH variations across the entire pH timeseries (Fig. 3). While
continuous recalibration of the optode was not possible, a two-point calibration, supplemented by
a third-point correction using a CRM, was performed prior to deployment (see Section 2.3).
Additionally, discrete underway TA and DIC samples, collected twice daily when possible, were
used to assess and correct for drift in the optode measurements.
The manufacturer specifies a drift of <0.005 pH units per day at 25 °C, though drift may vary
slightly with temperature. During our deployments, seawater temperatures ranged from 13.4°C to
22.0 °C in the North Atlantic and from 19.8°C to 27.7 °C in the South Pacific. While the cooler
North Atlantic conditions may have reduced drift rates slightly, temperatures during the South
Pacific cruise were close to or above 25 °C for extended periods, and full manufacturer-specified
drift is likely to have occurred. Even if drift remained at the nominal rate of 0.005 pH units per
day, this would amount to a cumulative offset of up to 0.175 pH units over ~5 weeks (SO279) and
0.28 pH units over ~8 weeks (SO289), in line with the deviation observed in our raw data (Fig. 3).
Notably, even with a recalibration during the cruise, drift before and after that point would still
introduce offsets. Thus, over timescales of days and longer, the accuracy of the measurement is
dependent on the correction to the TA and DIC samples.

To further assess the robustness of our drift correction, we compared our optode-corrected pH and
derived fCO$_2$ values with independent continuous measurements from autonomous sensors (Figs.
S1 and S2). The corrected optode pH exhibited a lower mean deviation (~0.008) from discrete pH
(TA/DIC) measurements than the independent spectrophotometric SAMI sensor (~0.02 pH units),
indicating that our correction effectively minimized drift-related inaccuracies (Fig. S1). The
overall scatter between the optode-corrected and SAMI-measured pH was modest (RMSD =
0.0329), suggesting reasonable consistency despite inherent sensor measurement noise. Similarly,
calculated fCO$_2$ from corrected optode pH showed a larger mean deviation (~12.7 µatm) compared
to directly measured fCO$_2$ (~5 µatm; Fig. S2). The relatively higher RMSD (30.76 µatm) reflects
variability in calculated fCO$_2$ arising from uncertainties in alkalinity estimates and carbonate
system calculations rather than fundamental flaws in the optode pH correction itself (Fig. S2).
Thus, these validations collectively confirm that our corrected pH data represent a reliable
improvement over raw optode measurements, suitable for robust biogeochemical analysis.

The overall post-cruise correction of pH involved an adjustment of approximately 0.4 to bring the
continuous measurements in line with discrete carbonate system observations. However, despite
this magnitude of correction, the internal variability within each cycle remains robust, with
observed diel fluctuations consistently within ~0.01. This fine-scale variability aligns with
expected temperature-driven pH changes and is distinguishable from random noise. The
measurement uncertainty of ~ ±0.01 means that while some small-scale variations approach the
uncertainty threshold, the structured nature of the observed diel trends—rather than random
scatter—supports their validity. If these variations were purely noise, we would not expect to see
systematic agreement with temperature fluctuations across multiple cycles and regions. To further
assess the robustness of these signals, we conducted a signal-to-noise ratio (SNR) analysis (see
Supplementary Information). SNR values exceeded 1 when pH was modeled based on temperature
(and salinity) in the North Atlantic, indicating that observed diel variability surpassed
measurement noise and reflected true environmental signals. In contrast, SNR values remained
below 1 in the South Pacific, suggesting that pH variability there was closer to the uncertainty
threshold and less clearly distinguishable from noise.
**2.6 Estimation of other biogeochemical parameters**
TA was estimated for the continuous pH dataset using the empirical equations presented by Lee et
al. (2006). For the North Atlantic, the corresponding equation was used (see Fig. S3 in
supplementary information):

$$TA_{NA} = 2305 + 53.97(SSS - 35) + 2.74(SSS - 35)^2 - 1.16(SST - 20) \\ - 0.040(SST - 20)^2 \tag{1}$$

In contrast, the (Sub)tropics equation (SBT, Eq. 2) was applied for the South Pacific region, as the
cruise mostly followed the 32.5°S longitude and this equation was determined to offer the best fit
to the local temperature (SST) and salinity (SSS) equation (see Fig. S3 in supplementary
information):

$$TA_{SBT} = 2305 + 58.66(SSS - 35) + 2.32(SSS - 35)^2 - 1.41(SST - 20) \\ + 0.040(SST - 20)^2 \tag{2}$$

TA estimates were used together with pH to solve the rest of the marine carbonate system (i.e. DIC
and $f$CO$_2$) using PyCO2SYS (Version 1.8.2; Humphreys et al., 2022), with the carbonic acid
dissociation constants of Lueker et al. (2000), the bisulfate dissociation constant of Dickson
(1990), the total boron to chlorinity ratio of Uppström (1974), and the hydrogen fluoride
dissociation constant of Dickson and Riley (1979).
While direct TA measurements were collected twice daily, their limited temporal resolution made
them unsuitable for continuous carbonate system calculations. The empirical TA equations from
Lee et al. (2006) provided a high-resolution dataset that allowed for more comprehensive system
reconstructions. A comparison of measured and estimated TA values (see Fig. S3 in supplementary
information) shows good agreement, with deviations generally within the uncertainty of carbonate
system calculations.

**2.7 Projected pH variability**

The derived parameters $pH_{temp}$, $pH_{sal}$, and $pH_{temp,sal}$ were calculated while holding TA and DIC constant at the average values of each diel cycle, where TA was estimated following Lee et al. (2006) and DIC was derived from measured underway pH and estimated TA. This approach allowed us to assess the expected pH response to changes in temperature and salinity alone, without introducing additional assumptions about concurrent variations in carbonate chemistry.

Similarly, $pH_{TA,fCO2}$ was computed while holding TA and $fCO_2$ constant at their daily mean values. $fCO_2$ was derived from measured pH and estimated TA, ensuring that the calculation reflects equilibrium conditions for given waters with distinct temperature-salinity properties while allowing temperature and salinity to vary independently. This formulation enables direct comparisons of observed pH to expected values under different scenarios, helping to disentangle the relative influences of abiotic drivers (temperature, salinity) versus processes such as air-sea $CO_2$ exchange and biological activity.

Our approach maintains consistency by using a single set of carbonate chemistry parameters (TA and DIC) as the baseline for assessing temperature and salinity influences. While pH is initially used to estimate DIC, the subsequent calculations isolate the effects of temperature and salinity without assuming variability in TA or DIC. This method enables direct comparisons between observed and expected pH, providing a clearer framework for distinguishing abiotic influences from biological processes and air-sea $CO_2$ exchange.

All calculations were done using the same configuration in PyCO2SYS described in Sect. 2.6.

**2.8 Identification of full diel cycles**

To account for the influence of geographic location on temporal measurements, timestamps from Coordinated Universal Time (UTC) were converted to Local Solar Time (LST). This conversion was necessary to align time-sensitive data with the true solar position at each measurement location, thereby facilitating more accurate comparisons of environmental data across different geographic regions and improving the analysis of diel processes.

The conversion process involved calculating the mean longitudinal position for each date within the dataset. Subsequently, a time offset was determined based on the average longitude, assuming a standard rate of Earth's rotation. This offset was then applied to the original UTC timestamps, resulting in a modified dataset with timestamps adjusted to reflect LST:

$$\text{DateTime}_{LST} = \text{DateTime}_{UTC} + \left( \frac{\text{Longitude}_{mean}}{15} \right). \tag{3}$$

Following the conversion to LST, the dataset was further processed to isolate complete diel cycles to ensure that only data representing full 24-hour cycles were included. The North Atlantic dataset included 7 complete diel cycles, while the South Pacific dataset included 11 complete diel cycles.

**2.9 Uncertainty propagation**
**2.9.1 Uncertainty in pH measurements and post-correction**
To provide a comprehensive uncertainty estimate, we considered multiple sources of potential
error. The manufacturer-reported accuracy (±0.05 for pH 7.5–9.0, ±0.1 for pH 7.0–7.5) represents
a systematic bias that was addressed through calibration against discrete CRM samples and is not
appropriate for inclusion as a random uncertainty. Likewise, the reported precision (±0.003 at pH
8.0) reflects the repeatability of the measurements but does not quantify the full range of
uncertainty for error propagation. The final uncertainty in pH measurements was estimated by
combining uncertainties from two main sources: (1) the uncertainty in the TA and DIC
measurements used to calculate $pH_{obs}$ and (2) the correction of the UWS pH measurements using
$pH_{obs}$
First, the precision in TA and DIC were determined based on the RMSE from repeated
measurements of a known standard water sample in the laboratory (NIOZ; 0.92 umol/kg and 1.95
μmol/kg, respectively). Then a Monte Carlo simulation was applied to the calculated $pH_{obs}$ to
obtain a $pH_{RMSE}$ for each subsample $pH_{obs}$.
Next, the pH measurements obtained from the optode were corrected using $pH_{(TA, DIC)}$ from the
discrete measurements of TA and DIC. The uncertainty in the pH correction was assessed using a
bootstrapping approach (n=1000 iterations), where a fraction (50%) of the discrete samples was
randomly omitted in each iteration and the selected fraction of the discrete samples varied within
its own $pH_{RMSE}$.
The variation in each subsample's $pH_{obs}$ captured the likely variability in TA and DIC
measurements, while omitting different subsets of data allowed for the estimation of how sensitive
the pH correction is to which set of subsamples are used for calibration.
**2.9.2 Uncertainty in pH diel patterns**
To ensure the observed patterns in pH over the full 24-hour cycles were not artifacts of sampling
bias or other anomalies, a Monte Carlo simulation (n = 1000 iterations) was employed on each diel
cycle's hourly data analysis. This simulation randomly selected 50% of the data points for each
hour (i.e. 50% of 120 hourly measurements for the North Atlantic dataset and 50% of 240 hourly
measurements for the South Pacific dataset), repeatedly calculating the mean pH to assess the
consistency and robustness of the hourly trends. This component of the uncertainty proved
insignificant (i.e., errors bars were smaller than the symbols in Fig. 4 and 5).
**2.10  $CO_2$ air-sea flux dynamics**
Air-Sea $CO_2$ fluxes were computed based on the relationship:

$$F = k_w \times K_0 \times (pCO_{2_{sw}} - pCO_{2_{air}})  \qquad (4)$$

where F represents the flux of $CO_2$ across the air-sea interface, $k_w$ is the gas transfer velocity, $K_0$
is the solubility constant and $pCO_{2_{sw}}$ is the partial pressure of $CO_2$ in sea water, representing the
concentration of dissolved $CO_2$ that is in equilibrium with the atmosphere and $pCO_{2_{air}}$ is the partial
pressure of $CO_2$ in the atmosphere above the ocean surface. For the fluxes, a positive value shows
the ocean acts as a source (i.e., releasing $CO_2$ to the atmosphere), while a negative value shows it
acts as a sink (i.e., absorbing $CO_2$ from the atmosphere). The parameterisation from Ho et al.
(2006) was used to determine the gas transfer velocity. All fluxes were computed using the
pySeaFlux package (v2.2.2; Fay et al., 2021).
Flux calculations were performed for each complete diel cycle, followed by the computation of
the mean flux for each cycle. Additionally, the mean flux for the diel cycle was also calculated
from the daily mean inputs (wind speed, temperature, salinity, and $p$$CO_2$) and specifically
computed for the hours 12 AM and 12 PM (LST) to examine temporal variations within each cycle
(Fig. 9).

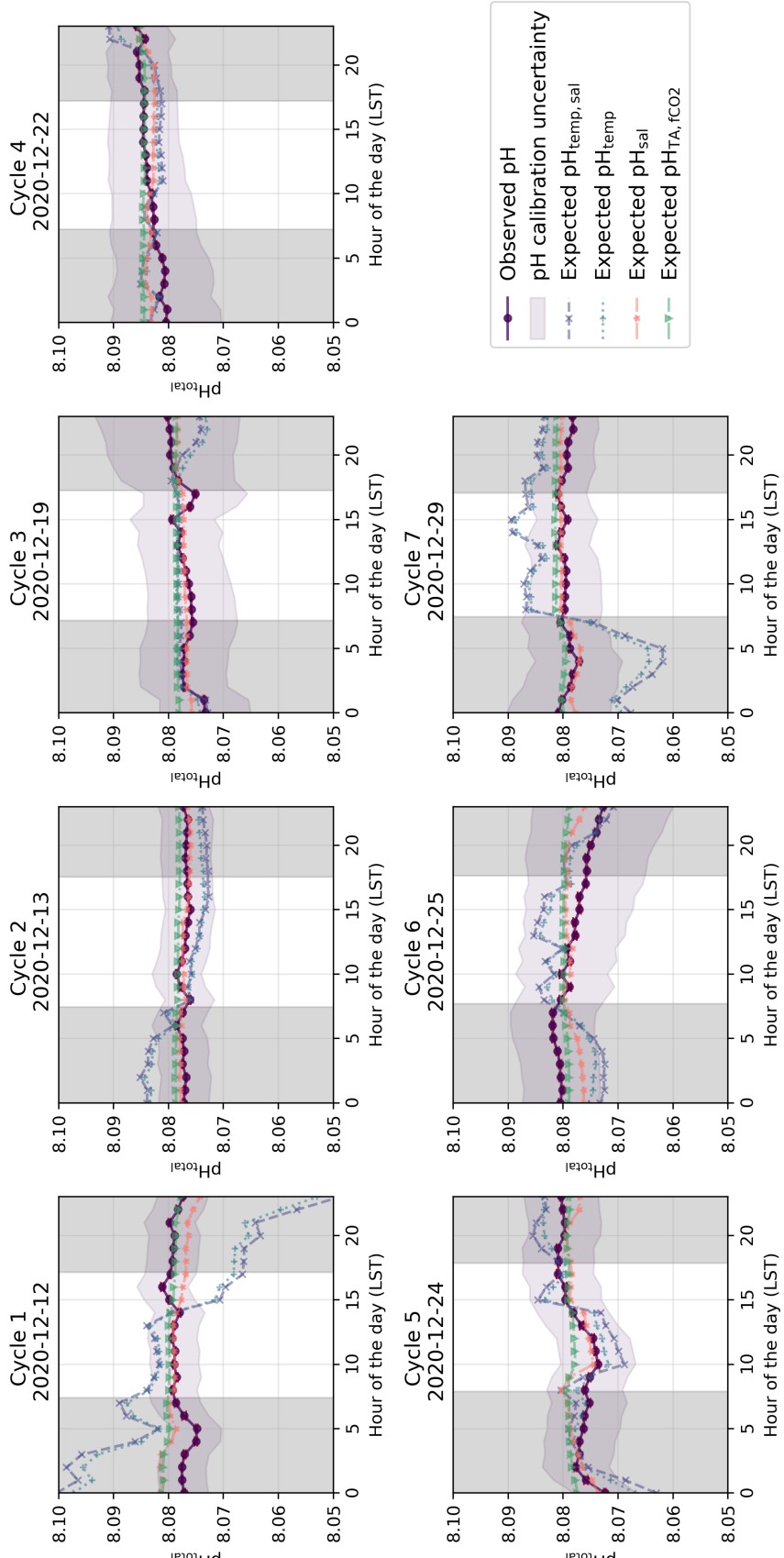

**Figure 4.  Identified diel cycles for cruise SO279 in the North Atlantic Ocean. Dark purple lines show observed pH (pH$_{obs}$), dashed grey lines show mean observed pH over the full diel cycle, and black lines show overall mean pH for all diel cycles combined for that cruise. The remaining shows expected pH using varying temperature and salinity (pH$_{temp,sal}$; dashed light purple lines), varying temperature alone (pH$_{temp}$; dashed blue lines), varying salinity alone (pH$_{sal}$; dashed orange lines) and constant TA and $f$CO$_2$ (pH$_{TA,fCO2}$; dashed green lines). Grey areas are night hours.**


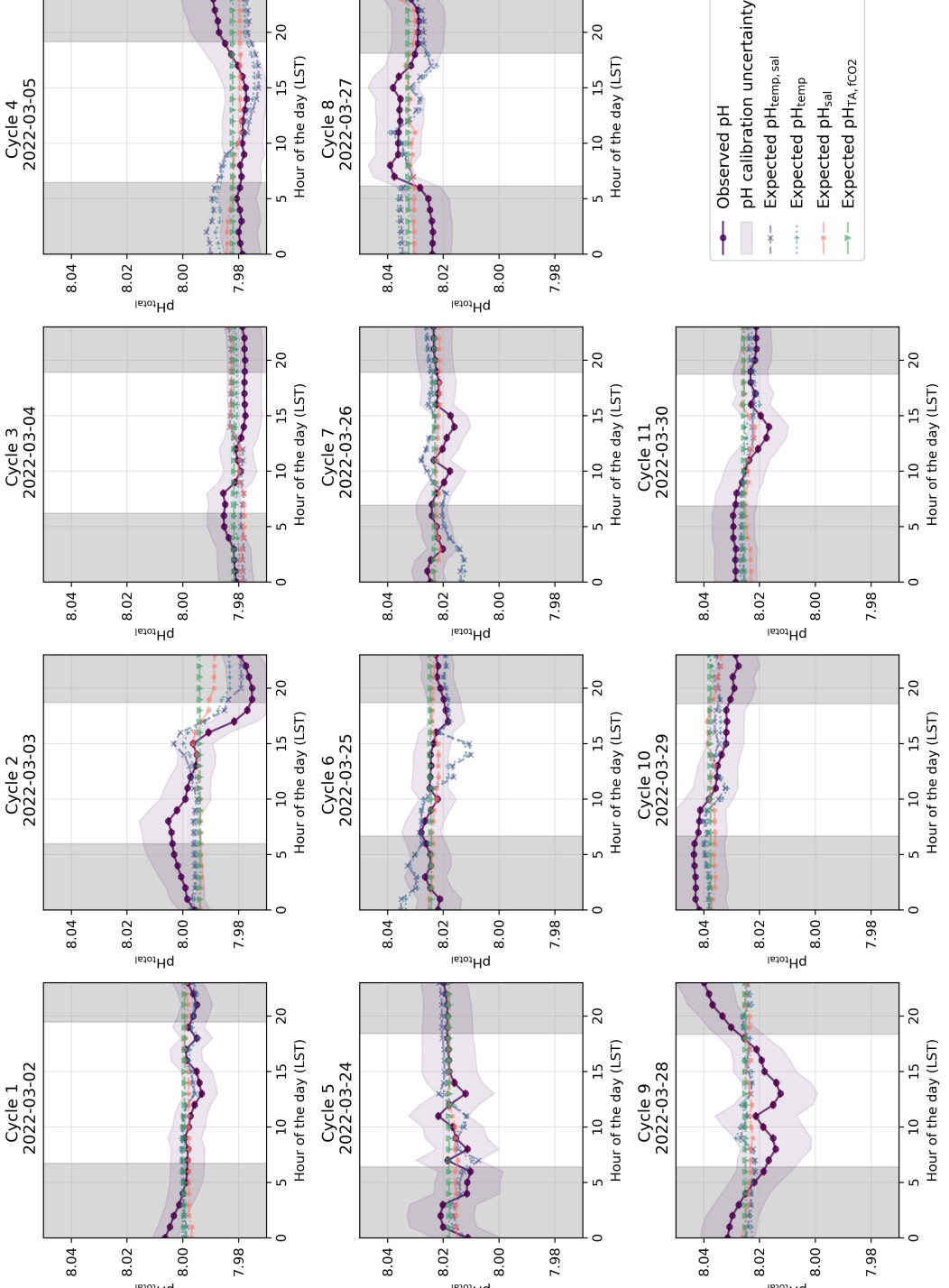

**Figure 4. Identified diel cycles for cruise SO289 in the South Pacific Ocean. Dark purple lines show observed pH (pH$_{obs}$), dashed grey lines show mean observed pH over the full diel cycle, and black lines show overall mean pH for all diel cycles combined for that cruise. The remaining shows expected pH using varying temperature and salinity (pH$_{temp,sal}$; dashed light purple lines), varying temperature alone (pH$_{temp}$; dashed blue lines), varying salinity alone (pH$_{sal}$; dashed orange lines) and constant TA and $f$CO$_2$ (pH$_{TA, fCO2}$; dashed green lines). Grey areas are night hours.**

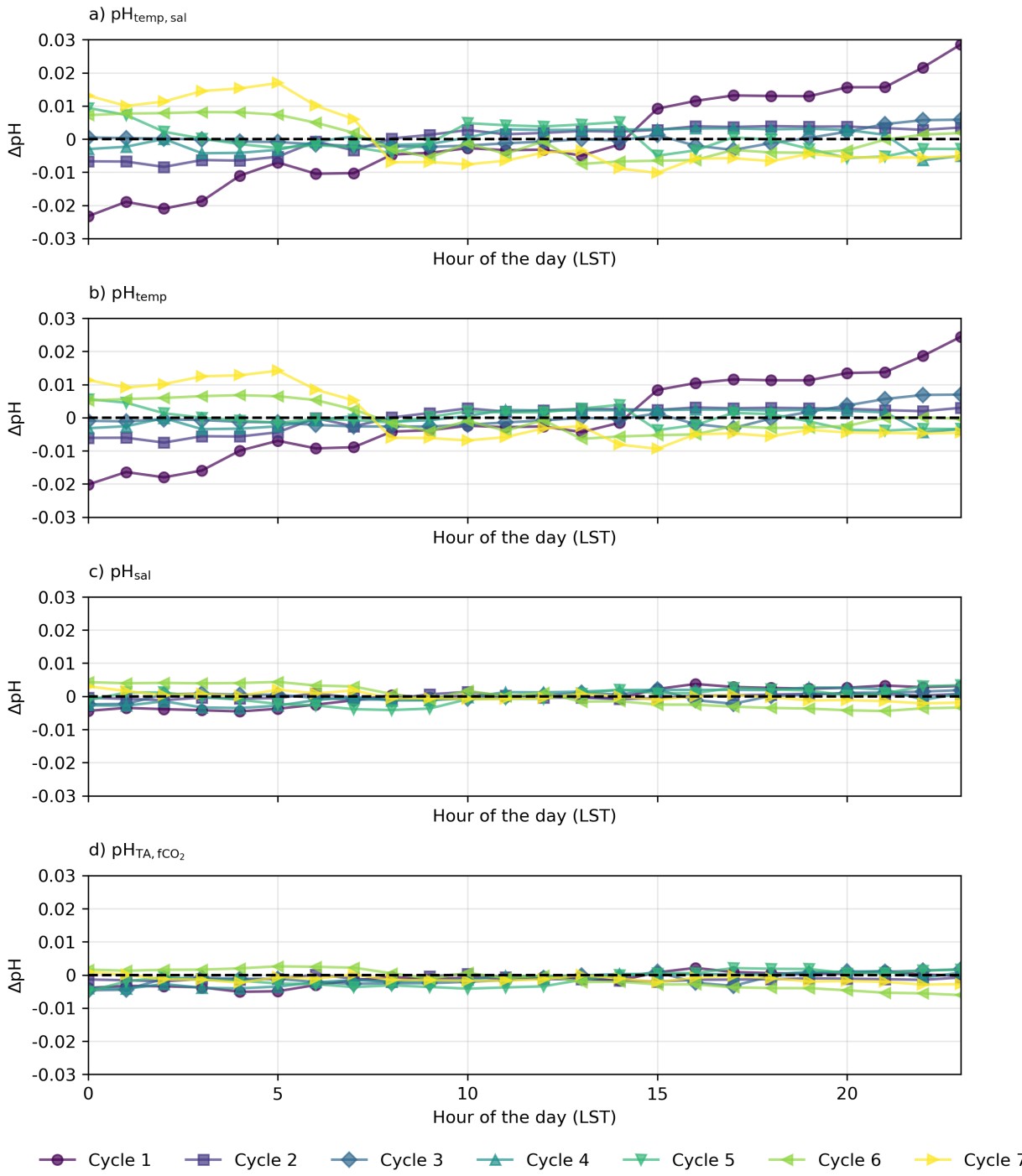

**Figure 6. Residual plots for diel cycles in the North Atlantic, illustrating the differences (ΔpH) between observed pH measured by the optode (pH$_{obs}$) and a) pH calculated from constant TA and DIC with varying temperature and salinity (pH$_{temp,sal}$), b) varying temperature only (pH$_{temp}$), c) varying salinity only (pH$_{sal}$) and d) constant TA and $f$CO$_2$ with varying temperature and salinity . Horizontal dashed lines at y=0 indicate no deviation between observed and calculated pH.**



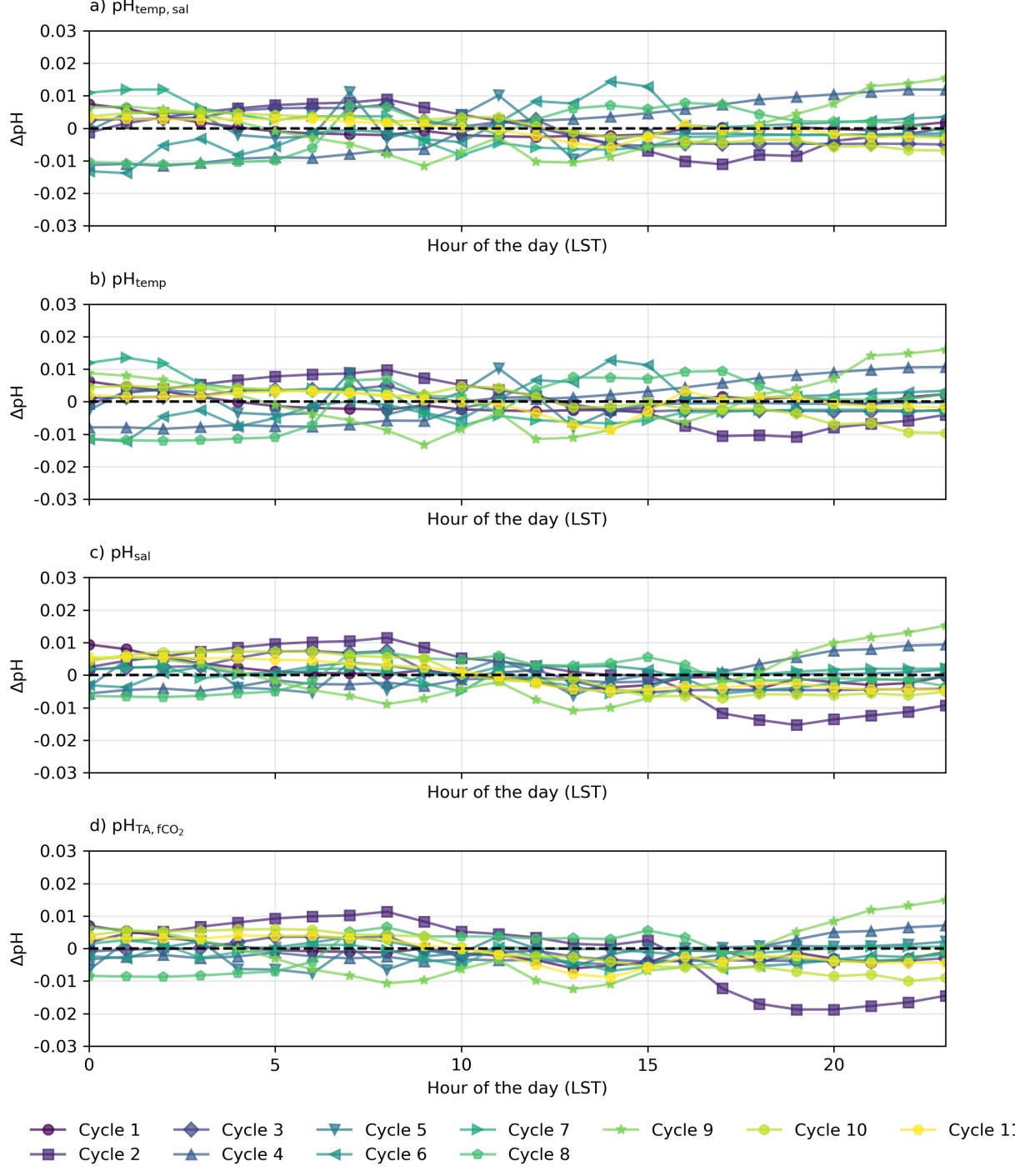

**Figure 7. Residual plots for diel cycles in the South Pacific, illustrating the differences (ΔpH) between**
**observed pH measured by the optode (pH$_{obs}$) and a) pH calculated from constant TA and DIC with varying**
**temperature and salinity (pH$_{temp,sal}$), b) varying temperature only (pH$_{temp}$), c) varying salinity only (pH$_{sal}$) and**
**d) constant TA and $f$CO$_2$ with varying temperature and salinity . Horizontal dashed lines at y=0 indicate no**
**deviation between observed and calculated pH.**

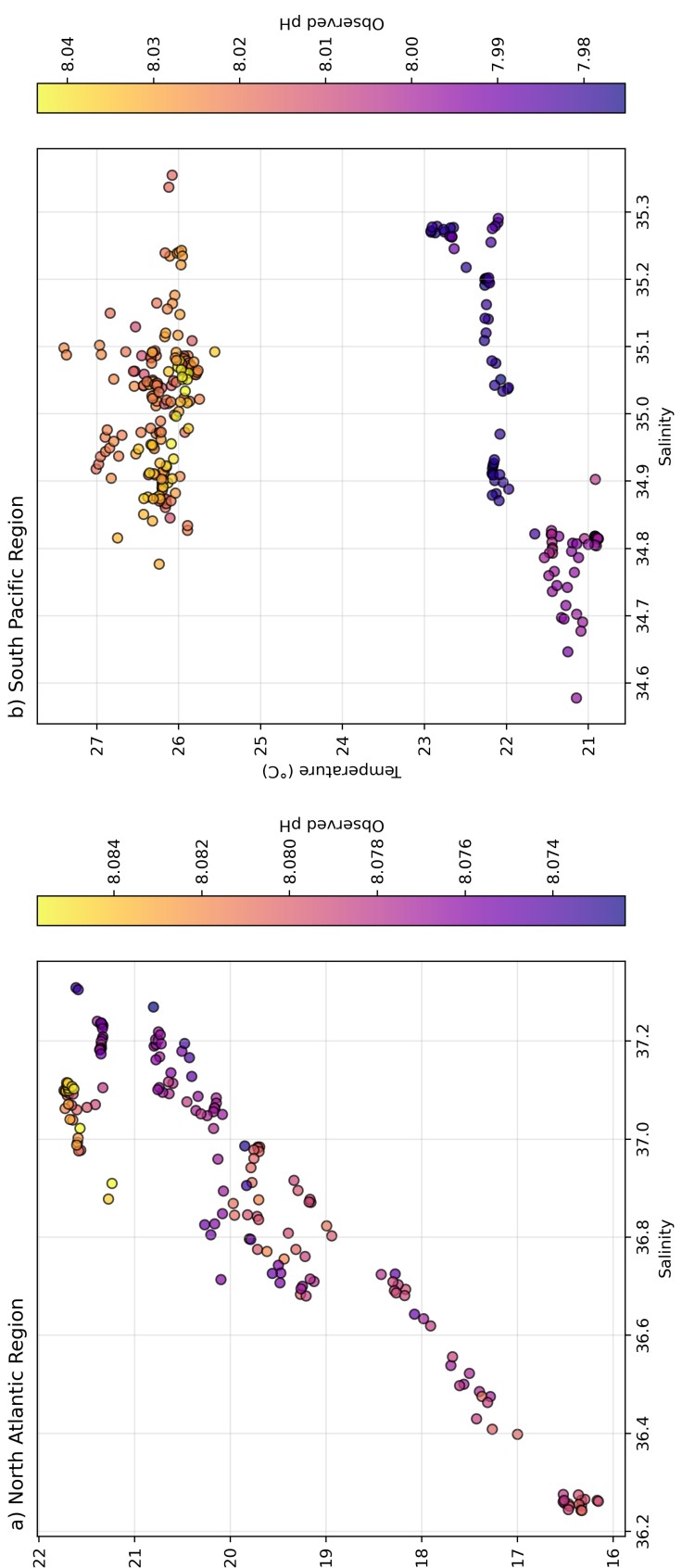

**Figure 8. T/S diagram with observed pH for a) the North Atlantic region and b) the South Pacific region.**

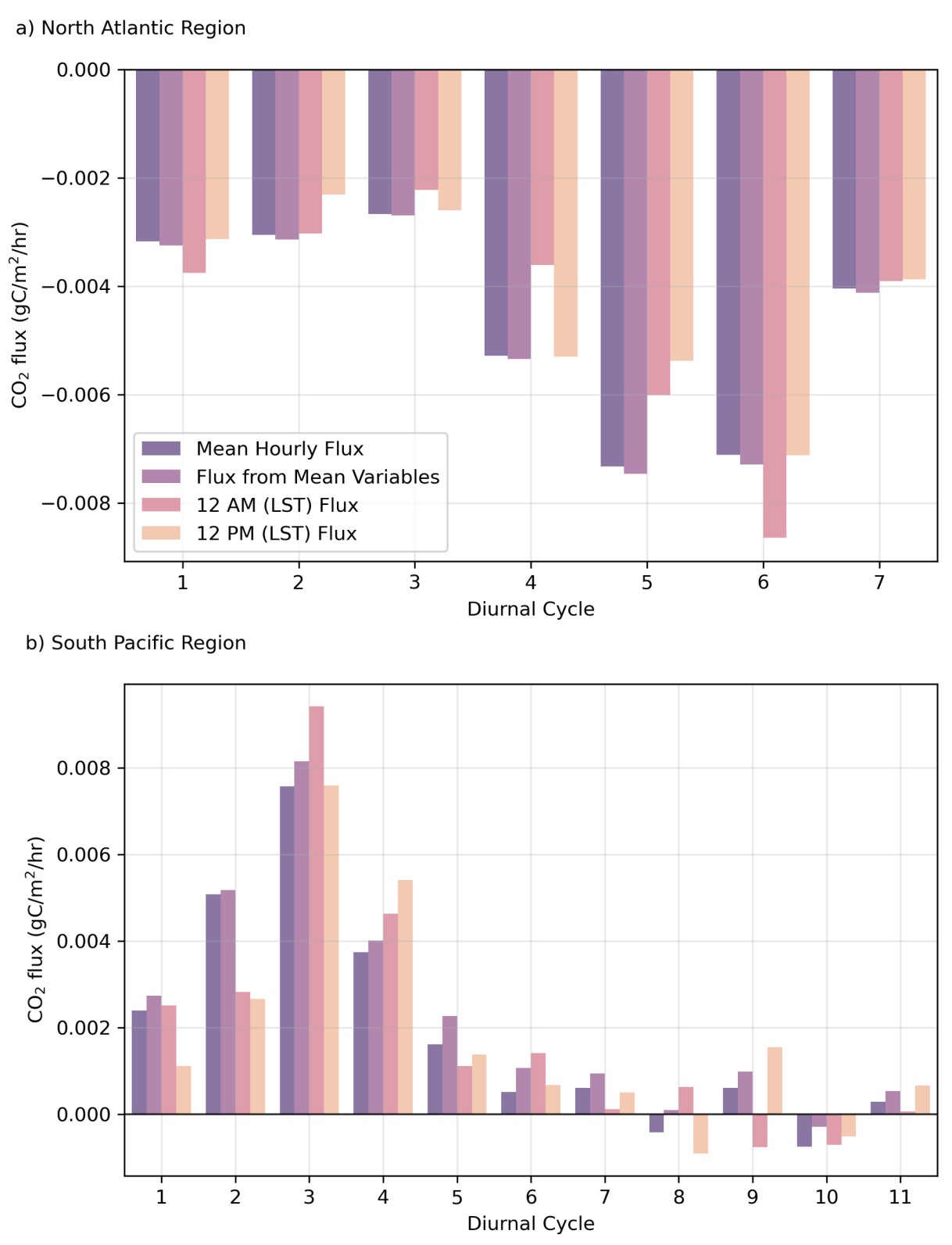

**Figure 9. Sensitivity analysis of mean CO₂ flux compared to flux calculated from mean inputs, fluxes for 12**
**AM (LST) and fluxes for 12 PM (LST). Top panel a) shows cruise SO279 in the North Atlantic while bottom**
**panel b) shows cruise SO289 in the South Pacific.**
**3 Results and Discussion**
We first analyse the effects of temperature and salinity on pH across different diel cycles and
regions (Sect. 3.1) using high-resolution pH data enabled by our novel optical measurement
system. We examine the pH expected from temperature and salinity variations alone ($pH_{temp,sal}$),
disregarding changes in TA or DIC. If $pH_{obs}$ (i.e. corrected underway pH measurements using
discrete TA and DIC subsamples) aligns with $pH_{temp,sal}$, it suggests that recent temperature
changes, such as day-night cycles, primarily control pH. In this context, "recent" is relative to the
air-sea $CO_2$ equilibration timescale, i.e., temperature change that happened recently enough that
its effect on pH has not been modified by subsequent gas exchange. Next, we assess the influence
of hydrographic variations on pH by considering $pH_{TA,fCO2}$, which accounts for constant $fCO_2$
instead of DIC alongside constant TA and varying temperature and salinity. Alignment of $pH_{obs}$
with $pH_{TA,fCO2}$ indicates that slower or long-ago processes control pH. For example, an observed
change in temperature may be due to spatial variability, with the ship passing through different
waters with distinct temperature-salinity properties that have had different temperatures for long
enough to reequilibrate with atmospheric $CO_2$. By leveraging the high-frequency resolution of our
measurement system, we then explore the role of biological activity and its interaction with abiotic
factors by looking at the discrepancies between $pH_{obs}$ and both $pH_{temp,sal}$ and $pH_{TA,fCO2}$ (Sect. 3.2).
Finally, we distinguish between temporal and spatial variability in our measurements (Sect. 3.3),
considering implications for air-sea $CO_2$ equilibration timescales at fine spatio-temporal scales
(Sect. 3.4).
**3.1 Influence of temperature and salinity**
**3.1.1 Basin-scale comparison**
The diel cycles of pH observed in the North Atlantic and South Pacific Oceans are significantly
influenced by temperature. In the North Atlantic, observed pH stays within ±0.01 of $pH_{temp,sal}$,
supporting the role of temperature and salinity in driving pH changes (Fig. 6). However, in the
South Pacific, where SNR values for temperature and salinity-driven pH changes remain below 1,
observed fluctuations may not exceed measurement noise, making it uncertain whether these
variations reflect true environmental signals or instrument variability (Fig. 7; Table S2 in
Supplementary Information). . Despite this, the systematic nature of diel fluctuations—seen across
multiple cycles and their correlation with expected temperature-driven trends—suggests they are
meaningful rather than random variability.
Salinity alone does not appear to strongly influence pH in any observed cycle. For both ocean
basins, $pH_{sal}$ typically remains close to the mean pH of each cycle, rather than impacting the
observed pH significantly. This consistency suggests that daily salinity variations do not exert a
primary influence on the observed pH (Fig. 4 and 5; Fig. 6c; Fig. 7c; Tables S1 and S2 in
Supplementary Information). Thus, the rest of this section focuses on the effect of temperature on
the observed pH.
In the North Atlantic, the observed variability in pH generally matches the calculated $pH_{temp}$ and
$pH_{temp,sal}$, with most cycles' residuals within ±0.01 (Fig. 4 and Fig. 6a and b, Cycles 2, 3, 4, 5 and
6). This agreement suggests that temperature and salinity together explain most of the observed
short-term variations. However, some cycles show more pronounced deviations. Cycle 3
demonstrates a particularly strong alignment between observed pH and expected $pH_{temp,sal}$
throughout the day (mean residuals < 0.001; Fig. 4; Fig. 6a and b), while Cycle 4 also exhibits
minimal variation (residuals -0.007 to 0.003; Fig. 4).

In the South Pacific, pH variability follows a different pattern, While some cycles (eg. Cycles 1,
3, 4, and 6) show a similar agreement between observed pH and $pH_{temp,sal}$ (Fig. 5 ), overall, the
influence of temperature appears weaker than in the North Atlantic. The SNR analysis (see Text
S1 in Supplementary Information) confirms this, with North Atlantic temperature-driven pH
fluctuations exceeding noise (SNR = 1.39 for $pH_{temp,sal}$, 1.20 for $pH_{temp}$), while all pH variations in
the South Pacific remain below SNR = 1 (see Tables S1 and S2 in Supplementary Information).
This suggests that in the South Pacific, observed fluctuations may be more influenced by noise
than by real temperature-driven variability. However, the persistence of systematic diel
fluctuations suggests that meaningful signals are still present.

The physical oceanographic context of each basin likely contributes to these differences. In the
North Atlantic, significant mixing due to ocean currents, eddies, and upwelling introduces
substantial variability in temperature, salinity, and pH across different waters with distinct
temperature-salinity properties (Fig. 6a and b; see Fig. S6 in Supplementary Information). The
Gulf Stream and North Atlantic Drift contribute to this complexity, which may help maintain
stronger temperature-driven pH changes by continuously exposing surface waters to variable
thermal forcing (Liu & Tanhua, 2021). In contrast, the South Pacific exhibits more predictable
hydrographic dynamics, driven largely by advection and mixing within large-scale gyres and trade
wind systems (Vallis, 2017). This results in a more stable and homogeneous water column, where
observed pH fluctuations remain closer to the measurement uncertainty (Fig. 7).
**3.1.2 Diel pH variability**
Beyond the basin-scale differences, a key feature of the observed pH cycles is the variability
between daytime and nighttime conditions. In both basins, temperature changes between day and
night are expected to drive corresponding pH shifts due to the temperature dependence of the
carbonate system.

In the North Atlantic, observed pH generally follows expected temperature-driven changes, with
$pH_{temp}$ and $pH_{temp,sal}$ showing higher values at night and lower values during the day due to the
inverse relationship between temperature and pH (Fig. 4, Fig. 6). For example, in Cycles 3 and 6,
the residuals between observed and expected pH remain small throughout the day, reinforcing the
dominance of temperature as a control mechanism. However, in some cycles (e.g., Cycle 1),
nighttime pH remains elevated beyond what would be expected from temperature alone,
suggesting the influence of additional processes such as biological activity or air-sea $CO_2$
exchange.

In the South Pacific, similar diel patterns are present but exhibit greater variability across cycles.
Some cycles (e.g., Cycle 8) show clear daytime decreases and nighttime increases in pH, consistent
with temperature-driven changes (Fig. 5). However, in other cases (e.g., Cycle 9), observed pH
departs from the expected diel pattern, indicating that other factors may be influencing short-term
variability.

The persistence of nighttime pH anomalies in both basins raises questions about equilibration
timescales. While temperature-driven changes in pH and $f\mathrm{CO_2}$ occur rapidly in response to
solubility and speciation shifts, $\mathrm{CO_2}$ equilibration between the atmosphere and ocean occurs over
much longer timescales (weeks to months; Jones et al., 2014)). This means that pH adjustments
due to temperature occur almost instantly, but whether they persist over a full diel cycle depends
on the history of the hydrographic properties. If waters have recently equilibrated with the
atmosphere, its pH should closely follow $\mathrm{pH_{temp,sal}}$. However, if these waters have undergone rapid
temperature shifts without sufficient time for equilibration, observed pH may diverge from
temperature-based expectations. This highlights the interplay between rapid thermodynamic
effects and longer equilibration processes, particularly in regions like the North Atlantic, where
mixing introduces additional complexity.
Ultimately, our observations suggest that while temperature is a dominant driver of diel pH
changes, the extent to which these changes persist overnight depends on the physical
characteristics of the surface waters and their equilibration history. In the North Atlantic, where
mixing is more dynamic, the system appears to re-equilibrate more readily, whereas in the South
Pacific, longer equilibration timescales may contribute to greater deviations from expected pH
patterns.
**3.2 Influence of biological activity**
While temperature and salinity effects on pH have been addressed in Sect. 3.1, deviations from
expected patterns may be shaped by biological processes. The process of photosynthesis during
daylight consumes $\mathrm{CO_2}$, leading to a rise in pH, whereas respiration and decomposition at night,
which release $\mathrm{CO_2}$, lower pH (Falkowski, 1994; Falkowski & Raven, 2013). The balance between
photosynthesis and respiration hence affects pH and should result in a diel pH cycle. However,
biological signals in the open ocean are often weaker than in coastal or upwelling regions due to
lower phytoplankton biomass and primary production rates (Behrenfeld & Falkowski, 1997;
Johnson et al., 2010). As a result, strong diel biological pH signals are not typically expected in
oligotrophic oceanic waters. These biological processes can also influence the rate at which $\mathrm{CO_2}$
equilibrates between the ocean and atmosphere. For instance, intense photosynthetic activity
during daylight hours can rapidly deplete seawater $p\mathrm{CO_2}$ in surface waters, potentially accelerating
seawater $\mathrm{CO_2}$ uptake from the atmosphere. Conversely, nighttime respiration can increase $p\mathrm{CO_2}$,
slowing the outgassing process and thus extending the equilibration timescale. However, given an
equilibration timescale of months, these day and night changes likely get averaged out over longer
periods, resulting in an overall steady-state $\mathrm{CO_2}$ flux and pH when observed over longer temporal
scales.
These biological processes are particularly evident in some cycles from the North Atlantic where
the expected $\mathrm{pH_{temp,sal}}$ changes due to temperature and salinity do not align with the observed data
but do follow the distribution expected from biological activity (Fig. 4). ). However, the strength
of these pH variations should be interpreted with caution, as biological activity in open-ocean
regions tends to be relatively low (Longhurst, 2010). The North Atlantic subtropical gyre, for
instance, has low rates of primary productivity (Baines et al., 1994; Tilstone et al., 2003), which
suggests that biological-driven pH variations would be relatively small. This is especially evident
for Cycle 1, which shows an increase in pH during daylight hours and a decrease in pH during
night hours, with a peak in pH in late afternoon and the lowest pH occurring in the early morning
before sunrise (Fig. 4). Other cycles likely reflect the combined impact of biological processes and
temperature effects, as the observed pH does not fully align with the distribution expected from
any of temperature, salinity or biological activity (Fig. 4 and 6). In the South Pacific Ocean, some
cycles show significant variability within $pH_{obs}$ that are not mirrored by the expected $pH_{temp,sal}$
changes (Fig. 5, Cycles 2, 8). For example, Cycle 2 exhibits significant deviations in pH with
higher observed values in the morning and a drop in the afternoon (Fig. 5). Although this suggests
a potential biological influence, the SNR values remain below 1 across all expected pH estimates,
making it unclear whether the observed fluctuations are truly driven by biological activity or
simply fall within the instrument's noise threshold (Table S2 in Supplementary Information).
The interplay between photosynthesis during the day and respiration at night, with possible
contributions of other biological processes, may be behind the more pronounced peaks and troughs
of Cycles 2 and 9 (Fig. 6; Duarte and Agusti (1998). Nitrogen fixation consumes hydrogen ions,
increasing pH, whereas denitrification releases hydrogen ions, thereby decreasing pH (Richardson,
2000; Zehr & Kudela, 2011). However, without nutrient and oxygen data, we cannot directly
assess whether these processes also impacted the observed pH.
Although biological activity likely accounts for some variability, not all cycles exhibit the same
residual, presumably biotic pattern of variation (Fig. 4 and 5). This aligns with expectations, as
phytoplankton biomass and primary productivity in open-ocean regions can be highly variable,
and biological impacts on carbonate chemistry are typically more pronounced in coastal or
upwelling systems (Duarte & Agusti, 1998; Williams & Follows, 2003). Moreover, the SNR
results indicate that in the North Atlantic, temperature-driven processes dominate short-term pH
variability, while the contribution of biological activity remains unclear due to overlapping
influences (see Table S1 in Supplementary Information). In the South Pacific, where all signals
fall below the noise threshold (SNR < 1; Table S2 in Supplementary Information), the observed
pH fluctuations may be driven by measurement uncertainty rather than a clear biological or
temperature-driven signal.
This variability further complicates assessments of air-sea $CO_2$ equilibration, as localized
biological conditions may transiently alter $CO_2$ dynamics, influencing the timescale for reaching
equilibrium. This is more obvious in the South Pacific, where some cycles display more
pronounced night-time stability (Fig. 5, Cycles 3, 6, 7 and 11), while others have noticeable day-
time fluctuations that could align with photosynthetic activity, which typically increase during
daylight hours (Fig. 5, Cycle 8; Duarte & Agusti, 1998; Raven & Johnston, 1991). Cycle 8 in the
South Pacific shows a peak in observed pH around midday, likely due to increased photosynthesis,
followed by a decrease in the evening (Fig. 5). Other cycles do not show this pattern as clearly
(i.e., they even show a decrease of pH during the day), suggesting that, for some cycles, respiration
may dominate over photosynthesis also during daytime (Fig. 5). For example, Cycle 9 displays
significant variations in pH throughout the day, with notable decreases during daylight hours (Fig.
5). Additionally, as some cycles appear to conform to temperature-based pH expectations, the only
minor deviations observed suggest biological activity to be minor or represent a balanced
biological system where photosynthesis and respiration are in near-equilibrium (Fig. 5, Cycles 3
and 6).

Despite the strong impact of both abiotic and biotic factors on pH, some cycles exhibit fine-scale
trends that cannot be solely attributed to temperature fluctuations or biological activity. The fine-
scale trends observed, especially in Cycles 1 and 7 (Fig. 4), exceed what can be attributed to
temperature-induced changes alone and cannot be explained by biological activity, given the
atypically high carbon fixation rates required to explain the pH offsets (Basu & Mackey, 2018;
Wang et al., 2023). Indeed, the required biological carbon fixation would need to exceed 687 mg
$C m^{-3} day^{-1}$ to explain the offset (Text S2). Typical rates in open ocean waters are much lower,
generally ranging from 50 to 150 mg $C m^3 day^{-1}$ in nutrient-poor regions and can reach up to 1000
$C m^{-3} day^{-1}$ in upwelling zones during phytoplankton blooms, which is not the case here (Basu &
Mackey, 2018; Wang et al., 2023).
Comparing the impact of biological activity based on the offsets between $pH_{temp}$ and $pH_{obs}$ with
chlorophyll-a fluorescence data also shows no clear pattern (Fig. S4 and S5). This is expected as
fluorescence does not necessarily reflect instantaneous photosynthetic activity. Instead,
fluorescence primarily indicates the presence and abundance of phytoplankton. Therefore, while
fluorescence can provide insights into the overall biomass of phytoplankton, it does not directly
correlate with photosynthesis and respiration. The absence of a clear correlation between
fluorescence and the daily pH cycle, with some cycles even showing a decrease in pH during
daytime, confirms that the influence of waters with distinct temperature-salinity properties
important in shaping the local high-resolution pH profiles.
**3.3 Variability and stability in spatio-temporal pH patterns**
As examined in Sect. 3.1, considerable spatial variability is observed in the North Atlantic (Fig. 8
and Fig. S8), and this heterogeneity introduces complexity in deciphering the relative contributions
of abiotic and biotic factors to pH fluctuations (Gruber & Sarmiento, 2002). Residual plots for the
North Atlantic suggest that the observed discrepancies between $pH_{obs}$ and both $pH_{temp}$ and $pH_{temp,sal}$
are insignificant, indicating that variability in surface water characteristics—which include but are
not limited solely to temperature and salinity—do not dominate the observed pH variability (Fig.
6 and 8; Dumousseaud et al., 2010). The SNR results further confirm that temperature-driven pH
fluctuations are real signals in this region (SNR > 1; Table S1 in Supplementary Information),
supporting their role as key drivers of diel pH variability. Therefore, the hourly fluctuations in
$pH_{obs}$ and the observed discrepancies between $pH_{obs}$ and $pH_{temp/sal}$ across various cycles in this
ocean basin may not be attributed to pronounced spatial variability (Fig. 4). The T/S diagram also
demonstrates this spatial variability, showing a wide range of pH values correlated with differing
salinity and temperature profiles across waters with distinct temperature-salinity properties (Fig.
8), reinforcing that spatial dynamics do not significantly influence pH variation in this region. This
is also supported by the relative stability observed in Cycle 4, where the ship's consistent
positioning (i.e. on station) highlights the role of temporal variability rather than spatial variability
(Fig. S6). Although water could still be moving spatially around the ship, the variability could be
due to some degree to different waters rather than being limited to true in-situ temporal variability.
In contrast, the South Pacific cruise predominantly exhibits a stable transit through more
homogeneous surface waters as no SST gradient is observed, although it is not entirely free from
disruptions (Fig. 8 and Fig. S7; Qu and Lindstrom (2004). This suggests a limited role of
interactions between diverse surface waters. Despite the hourly fluctuations observed, all diel
cycles in the South Pacific tend to cluster closely around the mean pH for their respective 24-hour
periods, reflecting substantial stability within each cycle but considerable variability across
different days, particularly evident among consecutive cycles (Fig. 5). This underscores a clear
temporal delineation of diel cycles influenced by DIC changes (Fig. 5). However, the SNR results
indicate that all expected pH fluctuations in this region remain below the uncertainty threshold
(SNR < 1; Table S2 in Supplementary Information), making it difficult to determine whether
observed fluctuations reflect true environmental variability or are primarily within the
measurement uncertainty.
**3.4 Implications for air-sea $CO_2$ fluxes**
Our findings indicate that while temperature and salinity predominantly govern diel pH
fluctuations, additional variability arises from surface water dynamics and biological activity.
Temperature rapidly affects pH, but the slower rate of $CO_2$ equilibration with the atmosphere
moderates its impact on air-sea $CO_2$ fluxes. Although biological processes markedly influence
daily pH cycles, they do not fully account for the observed variability, especially in cases where
unusually high carbon fixation rates would be required to explain observed pH (Sect. 3.2). Notably,
the North Atlantic acted as a net sink of $CO_2$, whereas the South Pacific was a net source to the
atmosphere during the study period.
$CO_2$ fluxes were calculated for each complete diel cycle (Fig. 9). The flux calculations were
performed by computing the mean flux for each cycle from daily mean inputs (wind speed,
temperature, salinity, and $p\text{CO}_2$) and specifically for the hours 12 AM and 12 PM (LST) to examine
temporal variations within each cycle (Fig. 9). Despite different methods of calculation, the $CO_2$
fluxes remained relatively consistent, indicating that the variations in pH and thus $CO_2$ flux do not
significantly affect the air-sea $CO_2$ exchange in either basin.
This consistent result shows that, despite the processes influencing pH in these two ocean basins
during the study period, air-sea $CO_2$ exchange over longer timescales (e.g., months) can dampen
short-term pH variability, resulting in relatively low variability across short spatiotemporal scales
(kilometers/days).
**4. Conclusions**
High-resolution pH data enabled by our novel optical measurement system provides valuable
insights into the complex and variable nature of surface  ocean pH. Our observations in the North
Atlantic and South Pacific show that pH fluctuates on diel and hourly timescales, with variations
driven not only by temperature but also by the interplay between waters with distinct temperature-
salinity properties and biological activity. These factors do not operate in isolation, making it
difficult to attribute pH changes to a single dominant driver.
Although the processes governing pH variability are well-understood, our high-frequency
measurements demonstrate the challenge of disentangling their contributions at fine spatial and
temporal scales. This underscores the importance of continuous, high-frequency measurements,
which reveal the heterogeneity in surface pH that lower-resolution datasets might miss. In both
basins, the close correlation between $pH_{TA,fCO_2}$ and observed pH across diel cycles suggests that
air-sea $CO_2$ exchange plays a key role in stabilizing pH despite temperature fluctuations. The
observed pH stability implies that the ship primarily encountered waters that had already
equilibrated with atmospheric $CO_2$, reinforcing the idea that temperature-driven pH changes alone
are not always sufficient to explain variability in surface ocean carbonate chemistry.

While our results indicate that fine-scale variability in these regions was relatively subtle—and
may not significantly impact large-scale assessments—this insight could only be gained through
high-resolution observations. This approach not only reveals the complexity of pH regulation but
also provides valuable insight into the fine-scale physical and biological interactions that shape
surface ocean chemistry. Importantly, although lower-resolution datasets may be sufficient for
capturing broad-scale patterns in surface ocean $CO_2$ chemistry, our findings highlight the critical
role of high-frequency measurements in refining regional understanding and improving predictive
models of ocean acidification and air-sea gas exchange.
*Data availability.* The hydrographic and biogeochemical data presented here, together with the
processing code is freely available online at https://doi.org/10.5281/zenodo.15873817.
*Supplement.* The supplement related to this article is available online.
*Author contributions.* LD and MPH conceptualized the project. AM, CG, EPA, LD, LQ, MPH,
and YO curated the data. GJR, LD, and MPH performed the investigation. LD conceptualized the
methodology, used the necessary software, visualized the data, and prepared the original draft of
the paper. CG, EPA, GJR, LD, MPH, and YO reviewed and edited the paper.
*Competing interests.* The contact author has declared that neither they nor their co-authors have
any competing interests.
*Disclaimer.* Publisher's note: Copernicus Publications remains neutral with regard to jurisdictional
claims in published maps and institutional affiliations.
*Acknowledgements.* We are grateful to the officers and crew of the R/V *Sonne* and technical
support from GEOMAR for their support and assistance, and the opportunity to join cruises SO279
and SO289. Special acknowledgment goes to Paul Battermann who greatly assisted in the
sampling aboard the FS Sonne during SO289. Funding for cruise SO289 came from the BMBF
(grant 03G0289NA) to EPA. We thank Karel Bakker and Sharyn Ossebaar for their help in the lab
– this work would not be possible without them. LD also wishes to thank Sorbonne Université and
the Institut de la mer de Villefranche (France), in particular the OMTAB team, for hosting her
during the later stage of this research project, as well as Jean-Pierre Gattuso, Pierrick Lemasson
and Elsa Simon for the many brainstorming sessions. Figure 2 schematic made by cartographic
design - faculty of Geosciences - Utrecht University.

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
