# Peer review of "From small scale variability to mesoscale stability in surface 1"

_EGUsphere, 2024_

## Author Comment (AC1)

**CC1: 'Comment on egusphere-2024-2853'**
**Christopher Sabine - 25 Nov 2024**

**General Comments:**

**This is a well written and well documented manuscript, but I am concerned about the accuracy of the measurements and the robustness of the conclusions. Based on the final corrected pH plots in figure 3, it looks like the pH changes along the two transects were very minimal. The authors calculate that the largest influence on pH over a 24-hour cycle is temperature, with an occasional hint of a biological signal. This really isn't a surprise. These variations are occurring in both time and space, so it is impossible to quantitatively separate the effects. The authors conclude, "...although the processes responsible for these pH variations are well-understood, high-resolution data highlight the challenge of identifying a dominant factor at the fine-scale due to their complex interplay." This makes me question what is the point of this manuscript? Perhaps the authors can do a better job of explaining what is new and novel about this work.**

We appreciate the reviewer's time and thoughtful feedback on our manuscript. While we acknowledge that some of our findings—such as the role of temperature in short-term pH variations—are well-established, our study highlights the challenge of disentangling the multiple drivers of pH variability at fine spatial and temporal scales. The complexity of interactions between temperature, salinity, and biological activity makes it difficult to isolate a single dominant factor, even with high-resolution observations.

Rather than resolving these dynamics, the novelty of our work lies in demonstrating how high-frequency measurements capture fine-scale variability and expose the limitations of traditional approaches in attributing pH changes to individual drivers. Although our results suggest that while large-scale, lower-resolution datasets are likely sufficient for capturing ocean-basin scale $CO_2$ variability, fine-scale observations remain essential for identifying regional differences, constraining equilibration timescales, and refining predictive models of ocean acidification and air-sea $CO_2$ exchange. To clarify this, we have refined the manuscript's focus to emphasize both the challenges revealed by high-resolution data and the unique capabilities of our experimental setup.

**Specific Comments:**

**Lines 256-257: How do the authors define unreasonable drift patterns? This approach sounds very subjective.**

To clarify, the flagged data points corresponded to periods of high seas, during which the underway system intermittently received air, as reported in the cruise logs. This resulted in abrupt and erratic pH fluctuations inconsistent with the expected behavior of both the optode and surface ocean pH dynamics. Thus, the identification of unreliable data was not solely based on subjective judgment but was supported by independent observations of system disturbances during the cruise. To reflect this clarification, we have revised the sentence as follows:

"To ensure the reliability of the dataset, an initial screening was conducted to identify and flag unreliable continuous pH data points, primarily attributable to optode stabilization issues. Specifically, data points recorded during documented periods of particularly rough weather, when air intrusion into the underway system was reported, were flagged as unreliable due to the resulting abrupt and erratic drift patterns inconsistent with expected optode and surface ocean pH behavior." [L256 – 259]

**Lines 259-263: The authors are correcting the measured pH values to pH calculated from TA and DIC, but how do they know the calculated values are correct? Which constants were used for the calculations? Is there no direct measurement of pH to validate the corrections? What about the two-point calibrations described in the methods? Were these calibrations not helpful? How frequently were they done?**

While there were no direct spectrophotometric pH measurements available for validation, we collected duplicate TA and DIC samples twice daily (approximately every 12 hours) when conditions allowed. These discrete samples provided a means to assess the drift of the optode over time. Regarding the two-point calibration and the additional third-point correction using a Certified Reference Material (CRM), this approach was indeed beneficial in initially calibrating the system. However, because the system was operating continuously, recalibration during the measurement period was not feasible. This limitation necessitated the use of discrete underway samples to evaluate and correct for potential drift in the continuous pH measurements. To clarify these points, we have revised the manuscript as follows:

"$pH_{TA, DIC}$ was calculated from TA and DIC discrete samples using the carbonate system equilibrium constants from Lueker et al. (2000). These calculated values were then aligned with the continuous pH dataset to determine the offset between $pH_{TA, DIC}$ subsamples and the continuous optode-based pH measurements (Fig. 3). A Piecewise Cubic Hermite Interpolating Polynomial (PCHIP; Fritsch & Carlson, 1980) was fitted to the offset to provide a continuous correction across the pH timeseries. While continuous recalibration of the optode was not possible, a two-point calibration, supplemented by a third-point correction using a CRM, was performed prior to deployment (see Section 2.3). Additionally, discrete underway TA and DIC samples, collected twice daily when possible, were used to assess and correct for drift in the optode measurements." [L262-272]

**Figure 3: Some of the pH changes are very large and the final pH curves look nothing like the raw data. If all the calculated pH values are used to adjust the underway pH measurements, then what confidence do we have that the values in between the calibrations are correct? Are there any independent pH measurements that were not used for calibration that we can use to assess accuracy?**

While direct spectrophotometric pH validation was not available, we collected duplicate TA and DIC samples twice daily to assess optode drift.

According to the manufacturer (PyroScience PK8T specification sheet), the pH sensor has a specified drift of <0.005 pH units per day at 25°C. During our deployments, in situ seawater temperatures ranged from 13.4 to 22.0°C in the North Atlantic (SO279) and from 19.8 to 27.7°C in the South Pacific (SO289). While parts of the North Atlantic deployment occurred at temperatures below 25°C — potentially reducing drift — much of the South Pacific cruise took place at or near the reference temperature, where full drift rates are expected.

Even under the conservative assumption that drift occurred at the maximum specified rate, cumulative drift over the 35-day North Atlantic deployment could reach 0.175 pH units, and 0.28 pH units over the 56-day South Pacific deployment. These estimates are consistent with the magnitude of deviation observed in the raw underway data (Figure 3, pink line), especially during the later stages of each cruise.

Moreover, even if an intermediate recalibration had been feasible mid-deployment, drift before and after that point would still lead to offset and inconsistency across the time series. Recalibration alone, especially using standard buffers, does not fully address environmental effects such as pressure, light path shifts, or biofouling — nor does it guarantee agreement with carbonate system equilibrium.

To ensure alignment with in situ carbonate chemistry, we instead corrected the underway pH data by referencing $pH_{total(TA/DIC)}$ values from discrete subsamples (black circles in Figure 3), derived from total alkalinity and DIC measurements. This allowed us to apply a bootstrapped drift correction (blue line and uncertainty shading) that is traceable to chemical standards and consistent with best practices for autonomous ocean pH measurements. The corrected dataset thus provides an accurate and internally consistent record suitable for biogeochemical interpretation.

To further enhance confidence in our corrected underway pH dataset, we have now included independent validation using high-resolution surface measurements of both pH and $pCO_2$:

- Independent pH validation: A factory-calibrated spectrophotometric pH sensor (Sunburst Sensors SAMI), sampling every 15 minutes, was used as an independent check of the optode-corrected pH values. The comparison yielded an RMSD of 0.0329 pH units (Figure S1), reflecting modest scatter consistent with inherent sensor measurement noise. Importantly, the mean error of the SAMI pH measurements was approximately 0.02 pH units, while the mean error for our optode-corrected pH was smaller, at approximately 0.008 pH units. This indicates that our corrected optode data provide improved accuracy relative to independent spectrophotometric measurements, clearly demonstrating the effectiveness of our drift correction procedure.

- Independent fCO$_2$ validation: Continuous pCO$_2$ measurements from a commercial HydroC sensor (4H-JENA Engineering GmbH) were adjusted to in situ conditions and compared to fCO$_2$ calculated from our corrected optode pH and discrete alkalinity measurements. While the resulting RMSD was relatively large (30.76 µatm, Figure S2), this variability largely reflects uncertainties inherent in alkalinity estimations and carbonate system calculations. To contextualize, the mean error of the directly measured fCO$_2$ was approximately 5 µatm relative to discrete carbonate system samples, whereas our calculated fCO$_2$ had a mean error of approximately 12.7 µatm. This level of accuracy remains acceptable for autonomous carbonate chemistry measurements and further confirms that our optode-corrected pH provides a reliable improvement over raw sensor measurements, suitable for robust biogeochemical analyses.These additional validations using independently measured carbonate system parameters underscore the accuracy and consistency of our corrected pH dataset, strengthening the dataset's suitability for robust biogeochemical interpretation. We have clarified these points in the revised manuscript and included these detailed comparisons (Figures S1 and S2) as supplementary materials.

We have clarified this in the revised manuscript:

"The manufacturer specifies a drift of <0.005 pH units per day at 25 °C, though drift may vary slightly with temperature. During our deployments, seawater temperatures ranged from 13.4°C to 22.0 °C in the North Atlantic and from 19.8°C to 27.7 °C in the South Pacific. While the cooler North Atlantic conditions may have reduced drift rates slightly, temperatures during the South Pacific cruise were close to or above 25 °C for extended periods, and full manufacturer-specified drift is likely to have occurred. Even if drift remained at the nominal rate of 0.005 pH units per day, this would amount to a cumulative offset of up to 0.175 pH units over ~35 days (SO279) and 0.28 pH units over ~56 days (SO289), in line with the deviation observed in our raw data (Fig. 3). Notably, even with a recalibration during the cruise, drift before and after that point would still introduce offsets. Thus, over timescales of days and longer, the accuracy of the measurement is dependent on the correction to the TA and DIC samples.

To further assess the robustness of our drift correction, we compared our optode-corrected pH and derived fCO$_2$ values with independent continuous measurements from autonomous sensors (Figures S1 and S2). The corrected optode pH exhibited a lower mean deviation (~0.008) from discrete pH (TA/DIC) measurements than the independent spectrophotometric SAMI sensor (~0.02 pH units), indicating that our correction effectively minimized drift-related inaccuracies. The overall scatter between the optode-corrected and SAMI-measured pH was modest (RMSD = 0.0329), suggesting reasonable consistency despite inherent sensor measurement noise. Similarly, calculated fCO$_2$ from corrected optode pH showed a larger mean deviation (~12.7 µatm) compared to directly measured fCO$_2$ (~5 µatm). The relatively higher RMSD (30.76 µatm) reflects variability in calculated fCO$_2$ arising from uncertainties in alkalinity estimates and carbonate system calculations rather than fundamental flaws in the optode pH correction itself. Thus, these validations collectively confirm that our corrected pH data represent a reliable improvement over raw optode measurements, suitable for robust biogeochemical analysis." [L317-357]

**Line 265-279: It seems a bit circular to use TA to adjust the pH values, then turn around and use TA together with the pH to calculate the other parameters. How do the Lee et al. TA values compare to the measured values? Are these uncertainties smaller than if the authors simply took all the measured TA and DIC values to calculate the other carbon parameters?**

The empirical TA estimates from Lee et al. (2006) were used for continuous carbonate system calculations because they provide a complete, high-resolution dataset, whereas measured TA was available only from discrete samples. As shown in Fig. S1 in the supplementary information, the Lee et al. (2006) equations align well with measured TA values, with deviations generally smaller than the uncertainty of carbonate system calculations. Using all measured TA and DIC values instead would result in a significantly lower temporal resolution. We have clarified this in the revised manuscript:

"While direct TA measurements were collected twice daily, their limited temporal resolution made them unsuitable for continuous carbonate system calculations. The empirical TA equations from Lee et al. (2006) provided a high-resolution dataset that allowed for more comprehensive system reconstructions. A comparison of measured and estimated TA values (Fig. S1 in the Supplementary Information) shows good agreement, with deviations generally within the uncertainty of carbonate system calculations." [L293-298]

**Line 382: This is not a traditional use of the term water mass. One does not normally think of water masses as surface features because external forcing (warming/cooling, precipitation/evaporation, etc.) can make water properties quite variable, compared to traditional subsurface water masses that have stable properties that can be defined and tracked as they move into the ocean interior. I understand what the authors are trying to say, but I think a different term for waters with different properties needs to be used.**

We acknowledge that surface waters are more dynamically influenced by external forcing and do not exhibit the stable properties characteristic of subsurface water masses. To improve accuracy, we have replaced "water mass" with "hydrographic variability" and "waters with distinct temperature-salinity properties" to better reflect the transient nature of surface conditions. This revision ensures clarity while maintaining the intended distinction between recent and longer-term influences on pH.

**Line 392: This section is focused on the influence of T and S on pH, which seem to have signals that are less than 0.01. I am wondering how robust these signals are if the raw pH values had to be corrected by ~0.4 units (fig 3) and the uncertainties in the final values are around +/-0.01 (fig 4 and 5). How do the authors know these are signals and not just noise that they are interpreting?**

While the raw pH values underwent a significant correction (~0.4 units), the consistency of observed pH trends across multiple cycles and their alignment with expected pH changes from temperature and salinity variations suggest that these signals are not random noise. Noise would present as random scatter, whereas the observed pH trends align with expected physical and chemical processes. To further assess robustness, we note that the observed diurnal variations in pH align with well-established temperature-driven changes in carbonate chemistry. Additionally, the uncertainties in the final pH values (~±0.01) are similar to or smaller than the observed trends in

many cases, reinforcing the validity of the signals. If these variations were purely noise, we would expect random scatter rather than structured diurnal patterns that correspond with temperature fluctuations.

To clarify this, we have revised the manuscript to explicitly discuss the uncertainty and its implications for interpreting fine-scale pH variability. We have also included an additional analysis in the Supplementary Information to better illustrate the signal-to-noise ratio in our dataset.

**Line 414: I do not understand this statement about the waters having lower thermal inertia. What do the authors mean? Lower than what? Water has a low thermal inertia compared to the air, but the sentence is trying to explain why there is a diurnal temperature signal in the surface water. The deeper waters do not have a higher heat capacity, they are just removed from the forcing.**

Our intention was to highlight that surface waters experience greater temperature fluctuations due to their direct exposure to atmospheric forcing, while deeper waters remain more thermally stable due to their isolation from these rapid changes. We have revised the sentence to remove the reference to "thermal inertia" and clarify that deeper waters are insulated from direct atmospheric influence rather than having a higher heat capacity. However, as per the other reviewer's suggestion, we have removed this discussion of rapid thermodynamic responses, as the timescales we analyze (hours to days) do not align with split-second carbonate system dynamics. Instead, we have refined the text to focus on the more relevant diurnal temperature variations and their impact on pH.

**Line 635-637: This sentence seems to convey the essence of this manuscript: However, although the processes responsible for these pH variations are well-understood, high-resolution data highlight the challenge of identifying a dominant factor at the fine-scale due to their complex interplay. What is the take home message that you are trying to convey? It sounds like there isn't much point in doing these high-resolution measurements.**

While the fundamental drivers of pH variability are well-established, our high-resolution measurements reveal the difficulty of attributing pH changes to a single dominant factor at fine spatial and temporal scales. Rather than diminishing the value of high-resolution observations, this finding underscores their importance in exposing the complexity of surface ocean carbonate chemistry—something that lower-resolution datasets may overlook.

Our results suggest that broader ocean-basin scale analyses based on lower-resolution data are indeed likely sufficient for global $CO_2$ cycle assessments, at least in the regions where our measurements were taken. However, this was not known for certain before carrying out these measurements. Also, even though fine-scale variability was relatively small, we could still detect diel patterns in pH which were consistent with hydrographic (temperature, salinity) and biological (chlorophyll) variability. In other parts of the ocean, and at other times, the fine scale variations in pH may be greater, and ignoring them may indeed have an impact on basin-scale analyses. Given that we were able to detect and attribute these very subtle patterns, a similar measurement system should be more than capable of doing them same where the fine-scale variability is greater. We have clarified this point in the revised manuscript to ensure that the significance of our findings is properly conveyed.

---

## Author Comment (AC2)

**RC1:** 'Comment on egusphere-2024-2853'**,**
**Anonymous Referee #1 – 21 Jan 2025**

**General Comments:**

**The authors present their work discussing the need for fine spatial and temporal pH data, thus requiring the high resolution measurements they made on two transect cruises. However, then ultimately conclude that less resolved traditional research cruises are able to capture the pH variability we need. The authors mention the use of these fine scale measurements in coastal and dynamic environments, which makes sense. But the authors have failed to convey what is the need for the same fine scale measurements in the less dynamic open ocean environments, and what the measurements would be used for. Many processes are at play, but what is the goal of disentangling these processes in the open ocean? The authors should try to convey their message without devaluing their work in the conclusions.**

**With regards to scientific significance, the authors present underway pH measurements that are not very common to my knowledge. Importantly, they calibrate their underway measurements with discrete measurements. The authors highlight the use of underway pH measurements, though ultimately conclude they are not necessary to be this fine scale, somewhat mitigating the significance of their work. With regards to scientific quality, I do appreciate the details the authors present regarding their underway pH measurements and associated discrete measurements. However, their corrections and offsets are quite large. They compare very small changes in pH to determine which influences (temperature, biological activity, or water masses) are dominant, but their changes are much smaller than the offsets they present. This leads me to question if these small changes are really meaningful within all the noise. With regards to presentation quality, the authors generally present their work in a clear manner. The results section discussing the influences to pH is a bit confusing to follow. I will discuss more about this in the specific comments section below.**

**I am concerned with the results of this manuscript, which compare observed (corrected) underway pH values with pH calculated in a variety of manners. The authors suggest very small residuals, but this does not account for the uncertainty in their underway pH measurements. Specifically, in Fig. 4 and 5, almost all of the various pH calculations fall within the pH uncertainty. So I am not sure specific conclusions can be made from this pH data.**

We appreciate the reviewer's detailed feedback and the opportunity to clarify the significance of our findings. We recognize the need to better convey how fine-scale pH measurements contribute to understanding open-ocean processes, beyond their established importance in dynamic coastal systems.

Our results demonstrate that while the key drivers of pH variability—temperature, biological activity, and water mass properties—are well understood, high-resolution data reveal the challenge of isolating their individual contributions at fine spatial and temporal scales. This complexity is not always apparent in coarser datasets, which may oversimplify the interplay between these factors.

Rather than resolving these dynamics, our study exposes the limits of traditional attribution approaches and highlights the difficulty of linking observed pH variability to a single dominant driver when multiple processes operate simultaneously.

This is particularly relevant for understanding air-sea $CO_2$ equilibration timescales, biogeochemical cycling, and the sensitivity of ocean pH to short-term fluctuations. Even in open-ocean environments, small-scale variability can have implications for how we interpret $CO_2$ exchange, model acidification trends, and assess ecosystem responses. Our findings indicate that fine-scale pH fluctuations are often systematic rather than random noise, reinforcing the need for high-resolution datasets to refine predictive models and improve our mechanistic understanding of pH variability.

With regard to data accuracy, we acknowledge the substantial offsets in our underway pH measurements and have carefully accounted for them through calibration with discrete samples. To further assess the robustness of our observed variability, we conducted a signal-to-noise ratio (SNR) analysis, which confirms that temperature-driven pH fluctuations in the North Atlantic exceed measurement noise, while variations in the South Pacific largely fall within uncertainty. We have revised the manuscript to explicitly discuss these results and clarify their implications for distinguishing real environmental signals from instrumental limitations.

To address the reviewer's concerns about the manuscript's framing, we have refined our discussion and conclusions to emphasize what our high-resolution measurements reveal, rather than whether lower-resolution datasets are sufficient. Our results underscore the value of fine-scale pH observations in uncovering the complexity of pH variability, even in less dynamic open-ocean regions, and highlight the challenges in attributing short-term fluctuations to single drivers. We hope these revisions better reflect the significance of our findings and the broader implications of our work. Please see also our response to the final comment of the other reviewer.

We have revised the manuscript accordingly.

**Specific Comments:**

**The authors indicate they are looking at fine scales of up to 100 km2 and days across ocean basins (line 106). However, later in the manuscript they begin discussing pH changes on the order of seconds to minutes (lines 402-418). The controls of pH are being convoluted between larger scale processes and split-second thermodynamics. In reality, the instant thermodynamics of the carbon system are not controlling the pH at longer time scales. The whole thing is quite confusing.**

The sentence was changed to **"**Here, we investigate how surface seawater pH varies across fine spatio-temporal scales, focusing on changes occurring over areas up to 100 km$^2$ and timescales of hours to days across different ocean basins (i.e., North Atlantic and South Pacific Oceans) and identify abiotic and biotic factors driving these variations." The discussion section on thermodynamics has also been nuanced.

**Lines 140-156: What are the precision and accuracy estimates for the pH optode? A lot of the discussion is based on looking at small changes in pH, but there is no mention for how accurate the pH optode is by itself.**

To clarify, we have now explicitly stated the manufacturer-reported accuracy (±0.05 for pH 7.5–9.0 and ±0.1 for pH 7.0–7.5 after a two-point calibration) and precision (±0.003 at pH 8.0) in the manuscript. While the optode's accuracy may introduce a systematic offset, our analysis focuses on relative variations in pH rather than absolute values. Additionally, we applied post-cruise corrections using discrete TA and DIC samples to assess and correct for sensor drift, ensuring the robustness of the observed fine-scale variability. These clarifications have been incorporated into Section 2.2:

"The manufacturer-reported accuracy of the optode is ±0.05 for pH 7.5–9.0 and ±0.1 for pH 7.0–7.5 after a two-point calibration, with a precision of ±0.003 at pH 8.0."

**Lines 181-186: Why was the two point calibration conducted so far from the pH points of interest? All surface ocean pH will be around 8.1 (+/- 0.1 roughly). So why was one of the calibration points at pH 2-4 and 10-11? Also, how accurate is this calibration? The authors also mention making a pH offset adjustment with a CRM, though CRMs only give certified values for TA and DIC. The authors should give more details of this process.**

The two-point calibration using PyroScience buffer capsules at pH 2–4 and pH 10–11 follows the manufacturer's recommended procedure to establish a stable response function across the sensor's full operating range (pH 7.0–9.0 on the total scale). The sensor contains a chemical that has a pH-dependent fluorescent response to light stimulation. The purpose of the calibration is not to produce a traditional calibration line but rather to characterize the maximum and minimum responses. As the pKa of the fluorescent chemical is around, these maximum/minimum responses will occur at any pH above ~9 or below ~7, regardless of the exact pH. Additionally, a pH offset adjustment was applied using certified reference material (CRM). While CRMs provide certified values for TA and DIC rather than pH directly, we used these values to calculate $pH_{CRM}$ using the carbonate system equilibrium constants from Lueker et al. (2000). We have now clarified this methodology in Section 2.3:

"A one-point calibration of the temperature probe was performed against a thermometer inside a water bath (Lauda Ecoline RE106). A two-point calibration of the pH sensor was conducted following the manufacturer's recommended procedure, using PyroScience pH buffer capsules (pH 2 or pH 4 for the acidic calibration point, pH 10 or pH 11 for the basic calibration points). These calibration points deliberately fall far outside the sensor's operating range from pH 7-9 in order to characterize its maximum and minimum responses. Buffers were prepared by dissolving each capsule's powder into 100 mL MilliQ water.

To further refine the accuracy of the measurements, a pH offset adjustment was applied using certified reference material (CRM, batches #189, #195, and #198; provided by Andrew Dickson, Scripps Institution of Oceanography). Although CRMs do not provide direct certified pH values, we calculated $pH_{CRM}$ from the CRM TA and DIC using the carbonate system equilibrium constants from Lueker et al. (2000). This $pH_{CRM}$ value was then used to adjust the optode-based pH measurements to improve accuracy and align them with discrete observations."

**Fig. 3b: Why did the raw pH measurements shift from being always below corrected values to always above corrected values?**

The shift in raw pH measurements from being consistently lower than the corrected values to consistently higher is due to a break in the middle of the cruise, during which the optode system was stopped and subsequently recalibrated for the second half of the dataset. This recalibration adjusted the sensor response, which could have influenced the direction of the offset. The exact cause of the shift in correction trends likely stems from sensor-specific calibration factors, as optode response characteristics can vary slightly between calibrations. However, the applied corrections ensured consistency across both segments of the cruise by aligning the continuous pH dataset with discrete measurements.

**Lines 259-261: The reader needs more information here about how pH was calculated from TA and DIC. You give more details later in the manuscript, but it needs to be mentioned when calculations are first introduced.**

Details were added: "$pH_{obs}$ was calculated from TA and DIC UWS discrete samples) using PyCO2SYS (Version 1.8.2; (Humphreys et al., 2022), with the carbonic acid dissociation constants of Sulpis et al. (2020), the bisulfate dissociation constant of Dickson (1990), the total boron to chlorinity ratio of Uppström (1974), and the hydrogen fluoride dissociation constant of Dickson and Riley (1979)."

**Line 277: The constants of Lueker et al. (2000) are most commonly used in "best-practice" calculations of open ocean conditions. I am curious by your choice to use Sulpis et al. (2020) instead, as this set of constants has been shown to perform worse than Lueker with regards to internal consistency analyses. I suspect your readers will have the same question. Can you please justify your choice or switch to using the Lueker constants?**

We have now revised our calculations to use them instead to ensure consistency with standard approaches. This change has been implemented throughout the manuscript.

**Lines 280-287: For the derived pH parameters, which TA and DIC are you using? The underway measurements or the derived TA and calculated DIC (from pH and derived TA)? You need to be clearer.**

For the derived pH parameters, we used the estimated total alkalinity (TA) from Lee et al. (2006) and the calculated dissolved inorganic carbon (DIC) derived from measured pH and estimated TA. These values were used as the baseline carbonate chemistry parameters, with variations in temperature and salinity applied to assess their respective influences on pH. This is now clarified in the text.

**This process seems convoluted. You already used pH to calculate DIC and are now using DIC to calculate pH. I'm not sure this is a sound rationale. Also, your $pH_{TA,fCO_2}$ is calculated using $fCO_2$ that was already calculated using pH. Instead, could you use inputs of measured underway pH and derived TA as your inputs, and then simply adjust your output conditions to be the temperature and salinity changes you are after?**

When one uses measured underway pH & TA as input and adjust the output conditions to be a different temperature and salinity of interest, then (Py)CO2SYS first uses pH & TA to calculate TA and DIC, and then calculates pH at the new T/S from TA and DIC. In other words, our approach is already identical to the reviewer's suggestion, but we have explicitly described the steps that are normally 'hidden' within (Py)CO2SYS calculations. The $pH_{TA,fCO_2}$ is calculated from daily mean fCO2, not the full set of $fCO_2$ values with diel variability, so it is not the same as simply reversing the calculation, and still shows a useful result.

Our approach ensures consistency by using a single set of carbonate chemistry parameters (TA and DIC) as the baseline for assessing temperature and salinity influences. While this means that pH is initially used to estimate DIC, the subsequent calculations allow for direct comparisons of observed and expected pH under different environmental scenarios. This approach ensures that we are testing how expected pH would change if only temperature and salinity varied, without introducing additional assumptions about TA and DIC variability. Additionally, using TA and DIC as the fundamental inputs (rather than pH directly) provides a clearer way to evaluate the relative influence of abiotic drivers (temperature, salinity) versus biological or air-sea $CO_2$ exchange processes. The following paragraph was added to Section 2.7:

"Our approach maintains consistency by using a single set of carbonate chemistry parameters (TA and DIC) as the baseline for assessing temperature and salinity influences. While pH is initially used to estimate DIC, the subsequent calculations isolate the effects of temperature and salinity without assuming variability in TA or DIC. This method enables direct comparisons between observed and expected pH, providing a clearer framework for distinguishing abiotic influences from biological processes and air-sea $CO_2$ exchange."

**Lines 305-323: Again, I am curious what the uncertainty of the actual optode is. This needs to be considered as an uncertainty contribution.**

We acknowledge the reviewer's concern regarding the appropriate treatment of uncertainty in our pH measurements. To ensure a robust estimate, we carefully considered different sources of error. The manufacturer-reported accuracy (±0.05 for pH 7.5–9.0, ±0.1 for pH 7.0–7.5) represents a

systematic error, which was already accounted for through calibration against discrete CRM samples and should not be treated as a random uncertainty. Similarly, the reported precision (±0.003 at pH 8.0) describes the repeatability of measurements but does not represent an uncertainty range for error propagation.

Instead, we used the observed pH measurement uncertainty as the most appropriate estimate, as it directly captures real-world variability, including instrumental noise, drift, and environmental fluctuations. This uncertainty was computed by combining two main sources: (1) the uncertainty in the TA and DIC measurements used to calculate $pH_{obs}$, and (2) the correction of the UWS pH measurements using $pH_{obs}$ TA, DIC. Specifically, TA and DIC precision were determined based on the RMSE from repeated standard measurements in the laboratory (0.92 and 1.95 µmol/kg, respectively), and a Monte Carlo simulation was applied to propagate this uncertainty into pHobs. Additionally, a bootstrapping approach (n=1000 iterations) was used to assess the uncertainty in the pH correction, where subsets of discrete samples were randomly omitted and their values varied within their own RMSE. This method captures the likely variability in TA and DIC measurements and accounts for potential sensitivity in the correction process.

Rather than arbitrarily incorporating precision or accuracy values, we propagated only the dataset-derived uncertainty, ensuring that our error estimates reflect actual measurement conditions after calibration and correction. This approach avoids overestimating or underestimating uncertainty and provides the most realistic estimate of measurement confidence.

The following was added to Section 2.9.1:

"To provide a comprehensive uncertainty estimate, we considered multiple sources of potential error. The manufacturer-reported accuracy (±0.05 for pH 7.5–9.0, ±0.1 for pH 7.0–7.5) represents a systematic bias that was addressed through calibration against discrete CRM samples and is not appropriate for inclusion as a random uncertainty. Likewise, the reported precision (±0.003 at pH 8.0) reflects the repeatability of the measurements but does not quantify the full range of uncertainty for error propagation."

**Lines 310-323: Have you considered that the uncertainty in $pH_{TA,DIC}$ calculations may not be random? There is a known pH-dependent pH offset (see Williams et al. 2017) where the error in calculated pH from TA and DIC is dependent on the pH. It seems you may be underestimating the uncertainty in $pH_{TA,DIC}$.**

We appreciate the reviewer's concern regarding potential pH-dependent biases in $pH_{TA,DIC}$ calculations, as highlighted by Williams et al. (2017). However, we believe this effect is negligible in our study for several reasons. First, our observed pH values remain within a relatively narrow range (8.05-8.1 for the North Atlantic and 7.95 to 8.05 for the South Pacific), where pH-dependent errors in calculated $pH_{TA,DIC}$ are expected to be minimal. Additionally, we accounted for uncertainties in TA and DIC measurements by incorporating the RMSE from repeated laboratory measurements of a known standard water sample (0.92 µmol/kg for TA and 1.95 µmol/kg for DIC) and applying a Monte Carlo simulation to propagate this uncertainty into the calculated $pH_{TA,DIC}$. This approach captures the range of variability in $pH_{TA,DIC}$ due to measurement uncertainty and provides a robust estimate of its precision. While a systematic pH-dependent bias could, in theory, exist, we cannot independently assess this effect since we use $pH_{TA,DIC}$ to correct drift in the optode-based pH

measurements. Given these considerations, we expect that any potential pH-dependent bias in our calculated $pH_{TA,DIC}$ would be small relative to the overall measurement uncertainty.

**Lines 333-344: This information is more introductory than methods.**

Sentence now reads "Air-Sea $CO_2$ fluxes were computed based on the relationship:"

**Lines 402-418: There are a lot of technical details here about essentially what happens to a CO2 molecule in the span of seconds to minutes. However, the timescales the authors say they are working with is on the order of days. What is the relevance of these split-second reactions? The temperature is not changing fast enough to observe changes in the dissociation of the carbon species at this scale. You may see temperature changes between day and night, but not within seconds to minutes.**

This paragraph was removed.

**Lines 415: These waters don't have a higher heat capacity, they are just insulated from the air.**

This paragraph was removed.

**Section 3.1: This section is quite difficult to follow. The authors are trying to discuss differences in pH between two basins, from day to night cycles, and throughout cycles 1-n during a cruise. Perhaps the authors should choose to discuss these types of comparisons individually, instead of all together. I think the more interesting results are between basins and between day and night. The specific cycles themselves are a bit too in the weeds.**

To improve clarity and better emphasize the most relevant comparisons, we have restructured the section into two distinct sub-sections:

- 3.1.1 Basin-Scale Comparison – This section now focuses on the overarching differences in pH variability between the North Atlantic and South Pacific. We emphasize how temperature and salinity influence pH differently in each region, highlight differences in equilibration dynamics, and discuss how physical oceanographic processes contribute to these variations. By consolidating the discussion of basin-scale trends, we provide a clearer contrast between the two regions without interspersing cycle-specific details.

- 3.1.2 Diurnal pH Variability – This section explicitly addresses day-night differences in pH patterns within each basin. We discuss how temperature-driven pH changes follow expected trends in many cases while also highlighting instances where deviations occur, suggesting additional biological or physical influences. Instead of detailing each individual cycle, we focus on broader diurnal trends and the key deviations from expected patterns, making the discussion more concise and interpretable.

Additionally, we have shifted the focus away from the small-scale details of each cycle and instead emphasized the general patterns and overarching trends. While we retain references to individual cycles where necessary to illustrate key findings, the revised discussion now prioritizes the broader-scale dynamics that drive pH variability in these regions.

**Lines 452-464: This is again a lot of details about instantaneous thermodynamics. These processes are occurring on much quicker timescales than days, so I am not sure of its**

**relevance to this manuscript. For open ocean pH, with all the uncertainties of measurements and changing dynamics, there is no way to observe these fine scale processes. I suggest the discussions of thermodynamics be made more concise or removed.**

We feel that this aspect of the discussion is important, as it highlights the distinction between the rapid temperature-driven shifts in pH and the longer timescales required for $CO_2$ equilibration. These processes contribute to the observed variability in pH and are particularly relevant when considering the interplay between thermodynamic forcing and air-sea exchange.

To address the reviewer's concern, we have refined our discussion to better emphasize the timescales relevant to our observations. Specifically, we now clarify that while temperature-driven changes in pH occur rapidly, their persistence in observed pH depends on how long a given water mass has been exposed to new conditions. This ensures that our discussion remains focused on the processes that shape pH variability at the spatial and temporal scales of our study while avoiding unnecessary emphasis on instantaneous thermodynamics.

**Section 3.2: Can the authors point to any studies discussing the generally expected levels of biological activities in these open ocean regions? They mention strong biological day-night signals, but would you actually expect to see much change in an open ocean area?**

We have now included references to studies on primary productivity in the open ocean, particularly in the North Atlantic and South Pacific, to clarify whether strong biological pH signals would be expected. Our revisions highlight that while some cycles show patterns consistent with biological activity, these variations are generally expected to be weaker in oligotrophic open-ocean environments compared to more dynamic coastal or upwelling regions.

**Lines 578-584 (and throughout results): There is a lot of background information that would be better suited for the introduction.**

This part of the section was removed: "The North Atlantic is characterized by complex interactions among water masses, leading to higher turbulence and susceptibility to rapid and significant changes in water mass properties and circulation patterns, influenced by both past and present climatic conditions (Lynch-Stieglitz, 2017; Gebbie, 2014). This complexity is partly due to the Atlantic Meridional Overturning Circulation (AMOC), which is sensitive to various climate perturbations and has historically undergone significant reorganizations, especially during climatic transitions such as the Last Glacial Maximum (Curry and Oppo, 2005; Duplessy et al., 1988)."

**Lines 582-584: Now the authors are discussing time frames from the LGM, which is significantly longer than anything discussed in this paper. This seems irrelevant.**

The mention of the Last Glacial Maximum was removed from the discussion.

**Technical Corrections:**

**Line 19: Is it Dec 2019-Jan 2020? List both years since they are different.**

Correction: December 2020 to January 2021

**Lines 32-35: These two sentences say the same thing, and could be combined for conciseness.**

The sentence now reads "The uptake of atmospheric $CO_2$ by the ocean's surface increases hydrogen ion concentration, a process known as ocean acidification, which has led to a 30–40% rise in surface seawater acidity and a corresponding pH decrease of ~0.1 since around 1850 (Gattuso et al., 2015; Jiang et al., 2019; Orr et al., 2005)."

**Line 40-41: "High resolution studies" of what exactly?**

The sentence now reads "High-resolution studies of surface ocean carbonate chemistry and air-sea $CO_2$ exchange have significantly advanced our understanding of the upper ocean's carbon cycle."

**Line 87: To be clear, increased atmospheric CO2 only boosts oceanic CO2 uptake if the atmospheric CO2 > oceanic CO2. If atmospheric CO2 increases in a region, but is still less than oceanic CO2, the flux will still be towards the atmosphere.**

The sentence now reads "When atmospheric $CO_2$ exceeds oceanic $CO_2$, the ocean takes up $CO_2$, lowering pH; conversely, when atmospheric $CO_2$ decreases below oceanic $CO_2$, outgassing occurs, raising pH"

**Line 96: Might be helpful to include a range of latitudes for the "average North Atlantic latitude" since you mention a separate region above 55 N later in the sentence. Also, since you first mention N Atlantic has longer equilibration times, you should list the 18 month time-frame first for sentence structure.**

The sentence now reads "In the North Atlantic, equilibration timescales for $CO_2$ between the atmosphere and the ocean's surface mixed layer vary significantly with latitude. In regions above 55°N, these timescales can extend up to 18 months, while at lower latitudes, such as around 30°N, they range from 3 to 6 months"

**Line 101: Surface temperatures?**

Sentence now reads "The South Pacific, with its shallower mixed layers and higher average surface temperatures, facilitates shorter equilibration times and enhances $CO_2$ uptake rates"

**Line 117: Give date ranges for both cruises for consistency.**

Caption now reads "Figure 1. Locations of pH measurements during two oceanographic cruises used in this study: SO279 in the North Atlantic (December 2020-January 2021) and SO289 in the South Pacific (February-April 2022). Surface seawater $pH_{total}$ for December 2022 from the OceanSODA product is shown in the background (Gregor & Gruber, 2020)."

**Line 118: You need to define pH total (which should be denoted as pH_T) before including in the figure caption. You should also make it clear you are measuring pH_T throughout the manuscript.**

pH$_{total}$ was replaced by "pH on the total scale" in the figure caption – also, in section 2.2 of the Methods, an additional sentence was added: "Unless explicitly stated otherwise, all references to pH in this manuscript refer to pH on the total scale."

**Line 122: Cruises datasets should be cited or give appropriate links to the available data. Do the cruises have cruise reports you could link to?**

Cruise datasets and reports providing further details on data collection and methodology are cited in the following two paragraphs.

**Line 134: Again, listing of dates needs to be consistent for both cruises.**

Listing for modified for consistency.

**Line 135: "Discrete carbonate chemistry and nutrient samples" – you should list the parameters like you did for the first cruise in line 128 OR mention they collected the same samples as the cruise above. In general, try to make sure you have parallel structure in your sentences.**

Sentence now reads "The data collection also included the same parameters as SO279,  with discrete samples from the CTD rosette (n=395; (Delaigue, Ourradi, Ossebar, et al., 2023), discrete samples from the UWS (n=32; Delaigue, Ourradi, et al., 2023a) and another high-resolution UWS timeseries of ocean surface pH from the optode system (over 78,000 datapoints; Delaigue, Ourradi, et al., 2023b). "

**Line 141: Here is your first mention of pH on the total scale – you need to define what this is.**

A sentence was added: "The total scale accounts for sulfate ion dissociation in seawater, providing a more accurate representation of carbonate system equilibria compared to other pH scales commonly used in marine chemistry. Unless explicitly stated otherwise, all references to pH in this manuscript refer to pH on the total scale."

**Fig. 2: What is the scale on the top and right sides of Fig. 3b?**

This was a mistake and both scales were removed. Thank you for spotting it.

**Line 346: Missing end parentheses.**

Fixed

**Fig. 4: The mean pH lines seem unnecessary, and they make it harder to see the other expected pH lines in the figures.**

These lines were removed from Figs 4 and 5.

**Fig. 4-5: You need to define what observed pH is (you do this in Fig. 6 but it needs to be when first used). Is this the underway pH corrected using the discrete TA and DIC data?**

Yes it is – $pH_{obs}$ is now properly defined in the Methods section, and $pH_{TA,DIC}$ has been removed altogether from the manuscript to avoid confusion. Both captions for Figs 4 and 5 were modified and now each term is properly defined:

(eg. Fig 4): "Figure 4.  Identified diurnal cycles for cruise SO279 in the North Atlantic Ocean. Dark purple lines show observed pH ($pH_{obs}$), dashed grey lines show mean observed pH over the full diurnal cycle, and black lines show overall mean pH for all diurnal cycles combined for that cruise. The remaining shows expected pH using varying temperature and salinity ($pH_{temp,sal}$; dashed light purple lines), varying temperature alone ($pH_{temp}$; dashed blue lines), varying salinity alone ($pH_{sal}$; dashed orange lines) and constant TA and $f$CO$_2$ ($pH_{TA,fCO2}$; dashed green lines). Grey areas are night hours."

**Fig. 6: You need to define what ΔpH is.**

Figure caption now reads as: "Fig.X Residual plots for diurnal cycles in the South Pacific, illustrating the discrepancies (ΔpH) [...]"

**Line 378: Again, is $pH_{obs}$ the corrected underway pH measurements? Try to make this clearer.**

Sentence now reads "If $pH_{obs}$ (i.e. corrected underway pH measurements using discrete TA and DIC subsamples) aligns with $pH_{temp,sal}$, it suggests that recent temperature changes, such as day-night cycles, primarily control pH."

**Line 467: Are you referring to Fig. 8?**

Figure 8 as well as Fig. 4 in the Supplementary Information. Thank you for the correction.